# Quantifying contributions of chlorofluorocarbon banks to emissions and impacts on the ozone layer and climate

Megan Lickley [1]✉, Susan Solomon [1], Sarah Fletcher [2], Guus J.M. Velders [3], John Daniel[4], Matthew Rigby [5], Stephen A. Montzka [6], Lambert J.M. Kuijpers [7] & Kane Stone [1]

Chlorofluorocarbon (CFC) banks from uses such as air conditioners or foams can be emitted after global production stops. Recent reports of unexpected emissions of CFC-11 raise the need to better quantify releases from these banks, and associated impacts on ozone depletion and climate change. Here we develop a Bayesian probabilistic model for CFC-11, 12, and 113 banks and their emissions, incorporating the broadest range of constraints to date. We find that bank sizes of CFC-11 and CFC-12 are larger than recent international scientific assessments suggested, and can account for much of current estimated CFC-11 and 12 emissions (with the exception of increased CFC-11 emissions after 2012). Left unrecovered, these CFC banks could delay Antarctic ozone hole recovery by about six years and contribute 9 billion metric tonnes of equivalent $CO_2$ emission. Derived CFC-113 emissions are subject to uncertainty, but are much larger than expected, raising questions about its sources.

[1] Department of Earth, Atmospheric, and Planetary Sciences, Massachusetts Institute of Technology, Cambridge, MA 02139, USA. [2] Civil and Environmental Engineering, Massachusetts Institute of Technology, Cambridge, MA 02139-4307, USA. [3] National Institute for Public Health and the Environment (RIVM), 3720 Bilthoven, the Netherlands. [4] Earth System Research Laboratory, National Oceanic and Atmospheric Administrations, Boulder, CO 80305-3328, USA. [5] School of Chemistry, University of Bristol, Bristol BS8 1QU, UK. [6] Global Monitoring Division, Earth System Research Laboratory, National Oceanic and Atmospheric Administration, Boulder, CO 80305, USA. [7] A/gent b.v. Consultancy, Venlo, Netherlands. ✉email: mlickley@mit.edu

The Montreal Protocol to phase out production and consumption of ozone-depleting substances (ODS) has become one of the signature environmental success stories of the past century. Since entry into force in the late 1980s, the Protocol initiated global reductions and virtual cessation of new production of chlorofluorocarbons (CFCs) that dominate ozone depletion, first in developed and then developing nations, with all nations agreeing to essentially phase out production of CFC-11 and CFC-12 by 2010. Global actions have demonstrably avoided a world in which large ozone losses would have become widespread[1] and there are signs that the ozone layer is beginning to recover[2,3]. Because CFCs have lifetimes of many decades to centuries, atmospheric chlorine loading and ozone loss from these chemicals declines only slowly even after emissions cease. Further, CFCs were produced for use in equipment, some of which have lifetimes of up to multiple decades. This implies that a bank of material could still exist, contributing to current and future CFC emissions. Recent measurements of CFC-11 indicate that emissions of this gas have increased despite global reports of near-zero production since 2010[4,5]. This raises concerns regarding future ozone recovery[3] and how much emission could still be coming from banks stored in equipment. CFCs are also effective greenhouse gases, contributing to climate change. Indeed, the Montreal Protocol, while motivated by safeguarding the ozone layer, also reduced global warming that would otherwise have occurred (with about five times the equivalent greenhouse gas emission impact that had been anticipated from the Kyoto Protocol by 2010[6]).

A long-standing challenge in understanding the underlying causes of measured changes in ODS concentrations is in evaluating not just production and emission in a given year, but also the quantity of banked CFCs, subject to later release. In the 1970s, the majority of CFC emission was nearly immediate after production as most use was as spray can propellants, spray foam, and solvents, but as those uses were phased out, CFC use continued in applications designed to retain rather than release the material, such as refrigeration, air conditioning, and insulation foam blowing[7], increasing the bank of material that can leak out later. The observation of unexpected CFC-11 emissions after the 2010 global production phase-out[4] therefore highlights the need for the best possible understanding of how much CFC remains in banks worldwide and how much banks are contributing to current emissions and their changes over time. Continuing emissions from remaining banks are not prohibited under the Montreal Protocol, but recovery and destruction of unneeded CFC banks has been considered by policymakers as a means to both enhance ozone recovery and further safeguard the climate system as part of the Protocol[8]. The issue of additional production (potentially illegally or as an accidental by-product) is also a topic of scrutiny.

Previous work on evaluating banks focused on two primary methods, commonly referred to as top-down and bottom-up. In top-down analyses, bank magnitudes are obtained by the cumulative difference between global production (generally estimated from reported production values compiled by the United Nations Environment Programme (UNEP)) and emissions, estimated from observed mole fractions and an estimate of atmospheric destruction (a lifetime). Prior to 2006, this method had been the basis for international assessments of bank size, but is sensitive to small biases in some variables. In bottom-up analyses, an inventory of sales of material and leakage rates in different applications such as refrigeration, industrial processes, air conditioning, closed and open cell foams are carefully tallied and considered at different stages of application use[9]. Extensive bottom-up inventories for banks as reported in the Intergovernmental Panel on Climate Change's Technical and Economic Assesment Panel (IPCC/TEAP, 2005)[10] were much larger than

top-down estimates in the World Meteorological Organization (WMO) assessment of the time[7], raising important questions about why they differed and whether the benefits for ozone and climate of bank destruction policies might be greater than previously thought. A subsequent TEAP (2006) assessment[11] suggested that some of the discrepancy could stem from longer lifetimes, a result supported by later stratospheric modeling analysis[12]. Post-2006 WMO estimates adopted the bottom-up values for 2008 and integrated forward to diminish the influence of lifetime errors on derived bank magnitudes.

By using the broadest range of constraints to date in a Bayesian framework, we estimate that banks of CFC-11 and 12 are likely to be substantially larger than recent scientific assessments suggested[3], in part due to apparent underreporting of production. Current banks of these gases could delay ozone hole recovery by 6 years and contribute ~9 billion metric tonnes of equivalent $CO_2$ emission. Further, our analysis better quantifies key discrepancies between observationally derived emissions and reported production and emission values. Namely, we find that recent increases in CFC-11 emissions as well as ongoing CFC-113 emissions are considerably larger than expectations from banks and other sources, implying added unanticipated contributions to climate change and ozone depletion.

## Results

**Modeling framework.** Here we introduce a new Bayesian probabilistic approach to assess bank sizes and changes in emissions for the three primary chlorofluorocarbons CFC-11, 12, and 113. Observed mole fractions of each gas, together with lifetime scenarios, are used to infer emissions. We develop a process-based model using production and equipment information to construct Bayesian prior distributions for bank and emissions estimates (representing a bottom-up approach). Observationally derived emissions are then treated as observations in Bayes' Theorem and used to develop posterior estimates for the simulated emissions and banks. Posterior distributions therefore represent bank and emissions estimates in which observationally derived emissions are used to constrain uncertainties in bottom-up methods. We call this Bayesian Parameter Estimation (BPE; see Methods). This approach aids in understanding the differences between past evaluations using top-down and bottom-up methods. We also examine how current understanding of the atmospheric lifetimes of these gases propagates into bank sizes and uncertainties. Differences between annual production and sales (e.g., stockpiling) are possible but not included here due to lack of quantitative information. Our analysis suggests a substantial amount (up to 90% in the 1990s) of CFC-11 and 12 production has gone into banks, while CFC-113 provides a useful contrast, as it is not subject to significant banking. Continuing CFC-113 production for feedstock use remains substantial under the Montreal Protocol, but Parties are urged to minimize leakage. We examine how factors such as potential unreported production, uncertainties in bank release rates, and atmospheric lifetime assumptions affect BPE bank estimates. Here we address the following questions: What are best estimates and uncertainties in emissions of banked CFC-11, 12, and 113? How much could the bank from pre-2010 production contribute to recent increased emissions of CFC-11? How have emissions from banks likely contributed to delaying ozone recovery relative to a scenario where banks were recovered, and how much could they contribute to future delays if they are left unrecovered? Finally, how will bank emissions contribute to climate change if they are not efficiently recovered?

**Bayesian bank estimates and comparisons.** Figure 1a, b shows how top-down derived bank estimates provide large differences in

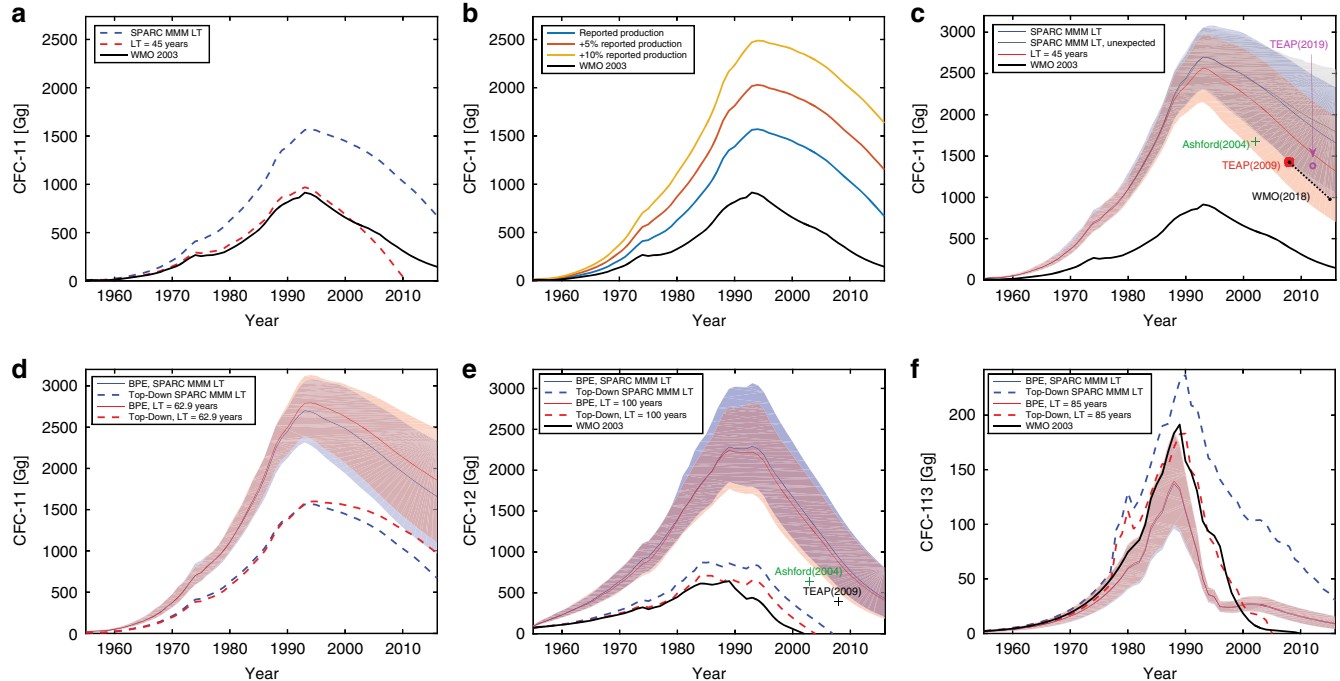

**Fig. 1 Bank estimates and comparisons.** Comparison of banks derived from Bayesian Parameter Estimation (BPE) along with previously published values, top-down bank estimates, and alternative assumptions. **a** Top-down CFC-11 bank estimates assuming known lifetimes and reported production (see Eq. 2). Banks are derived using SPARC multi-model mean (MMM) time-varying atmospheric lifetimes (blue) and a constant lifetime of 45 years (red). **b** Top-down CFC-11 bank estimates assuming SPARC MMM time-varying lifetimes and three production scenarios: Reported production (blue), 1.05× reported production (red), and 1.1× reported production (yellow). For (a) and (b) production values are based on AFEAS and UNEP reported values (see Methods). **c** BPE-derived CFC-11 bank estimates assuming the SPARC MMM lifetime (blue) and constant lifetime of 45 years (red). The gray line is analogous to the blue line but production prior includes additional production to account for unexpected emissions from 2000 to 2016 (see Methods). **d** BPE-derived CFC-11 bank estimates assuming SPARC MMM time-varying lifetimes (average value of 62.9 years) shown in blue, and constant lifetime of 62.9 years is shown in red. Dashed lines are corresponding top-down bank estimates. **e** BPE-derived CFC-12 bank estimate assuming SPARC MMM lifetimes (average value of 112.9 years) shown in blue, and 100-year lifetime is shown in red. Dashed lines are corresponding top-down bank estimates. **f** BPE-derived CFC-113 bank estimates assuming SPARC MMM lifetimes (average value of 106.3 years) shown in blue, and 80-year lifetime is shown in red. Dashed lines are corresponding top-down bank estimates. The black line in (a–c), (d) and (f) is the WMO (2003) bank estimate. For (c)–(f), the BPE median estimates are shown using thin solid lines with the 95% confidence intervals indicated by corresponding shaded region. The markers in plots (c) and (e) indicate previously published bank estimates as follows: the green marker is from Ashford (2004)[9], the red marker is from TEAP(2009)[32], the black marker is from WMO(2018)[3], where banks were derived beginning with TEAP(2009)[32] estimates, and the pink marker is from TEAP (2019)[33].

bank estimates for different lifetime and production assumptions for CFC-11. In Fig. 1a, we compare the top-down estimate for two different lifetime assumptions—first we consider a constant atmospheric lifetime of 45 years, taken from WMO (2003) estimates, and second we assume the time-dependent Stratosphere–troposphere Processes And their Role in Climate (SPARC) multi-model mean (MMM) lifetime scenario, which averages 62.9 years over the period considered (see Supplementary Fig. 1 and Methods). With the constant, shorter lifetime, the bank would have been fully depleted by 2010, whereas with the longer and time-dependent SPARC lifetimes a bank estimate close to 1000 Gg of CFC-11 is obtained in 2010. Both of these scenarios assume the same production prior over time, illustrating how the assumed lifetime scenario impacts the inferred banks with the top-down method. A comparison of these two lifetime scenarios is further discussed below.

Figure 1b illustrates the effect of a consistently larger production estimate on bank size. Here we assume the SPARC MMM lifetime scenario and allow production to be as reported, 5% larger than reported, or 10% larger than reported. Due to the cumulative effect of production on bank size, a 5% increase in production results in a bank size in the top-down approach that is ~50% larger in 2011, whereas a 10% increase in production results in a bank size that is ~100% larger. The two figures

underscore that the potential uncertainties in the banks are very large with the top-down approach.

None of the results shown in Fig. 1a, b make use of the uncertainties in observed CFC mole fractions, nor do they incorporate knowledge of the uncertainty ranges for a direct emissions factor (DE), release fractions (RF), or production, making it difficult to place any uncertainty on the results. Figure 1c, d shows the results of the BPE analysis for a range of assumed CFC-11 lifetimes (45 years, the SPARC MMM, and the time-averaged SPARC MMM of 62.9 years), and two production scenarios (one constructed with reported production, and one with additional unexpected production and emission starting in 2000; see Methods). While differences still exist between scenarios, the figure illustrates how uncertainties in the suite of inputs (including lifetime, production, observed concentrations, etc., see Eq. 4) better constrain the possible range in bank estimates compared to the fixed-input top-down approach.

Important factors in the differences in BPE bank size in the two atmospheric lifetime scenarios are the sensitivity to uncertainties in DE and RF. This is evident when comparing the posterior distributions of DE and RF for the two scenarios (shown in Supplementary Figs. 2 and 3). For the SPARC MMM lifetime scenario, both DE and RF posteriors are more noticeably skewed towards lower values from the prior distribution as compared to

the constant lifetime scenario. This suggests some key differences in the behavior of the posteriors between lifetime scenarios: the constant lifetime scenario of 45 yrs is associated with higher emissions leading to relatively larger DE values during high production years when the bank is still accumulating, and then relatively larger RF during low production years when a larger proportion of emissions is coming from the bank. This relationship is also illustrated in the joint posterior distributions of the banks with DE and RF, respectively (see Supplementary Figs. 4–7). Supplementary Fig. 5 confirms that the bank size is correlated most strongly with RF towards later time periods, and with DE (albeit only slightly) in earlier time periods (Supplementary Fig. 4). This strong negative correlation between bank size and RF in recent decades is to be expected for two reasons. First, for the simulation model, a low RF would lead to a larger accumulation in the banks in earlier decades. Because RF has high autocorrelation, a low RF in earlier decades would be correlated with a low RF in recent decades as well, thus explaining the strong negative correlation between RF and Banks in the prior. And second, for the posterior, if the near-zero reported production in recent decades is accurate, then emissions must be entirely driven by the depletion of the banks, and thus controlled by RF (i.e. Emissions $\cong$ RF × Bank). Therefore, we would expect values on the ridge where RF×Bank are closer to the observationally derived emissions to have a higher likelihood than values further from the ridge.

The prior and posterior production distributions are shown in Supplementary Fig. 8. The most noticeable difference in posteriors between the two lifetime scenarios occurs in the 1980s where the SPARC MMM lifetime results in a lower production posterior than the constant lifetime scenario of 45 yrs. Importantly, our posterior estimates of production indicate that total production from 1955 to 2016 of CFC-11 has likely been 13% (1-sigma $\cong$ 3%) larger than the values used in previous scientific assessments, contributing further to the discrepancies between the BPE bank size and WMO (2003) bank estimates.

An important result illustrated in Fig. 1 is that the BPE bank for CFC-11 is broadly consistent with the bottom-up bank estimates[9,10] with the BPE bank being the larger of the two. The great bulk of remaining CFC closed cell foams are thought to contain CFC-11, while remaining CFC in cooling systems is nearly all CFC-12 (this analysis and SROC[13]). Our analysis thus shows that the apparent contradiction between the bottom-up inventory assessment and the fixed-input top-down approach taken in scientific assessments up to the early 2000s can be reconciled when uncertainties are more extensively considered. It also implies that the total amount of material in the banks is indeed very likely to be much larger than thought by the best international WMO/UNEP scientific assessments in the late 1990s and early 2000s, both because of updated lifetimes estimates and a more extensive uncertainty analysis.

The impact of time-dependent lifetimes is shown in Fig. 1d with a comparison between the SPARC MMM and its average over the period considered of 62.9 years. The two scenarios produce similar bank sizes from 1955 to 1990, after which point the constant lifetime leads to a slightly larger bank size. This divergence is driven largely by the fact that the SPARC MMM lifetimes are decreasing throughout the time period such that prior to 1980, the SPARC MMM is larger than 62.9 years, and from 1981 onwards, it is smaller. In recent decades, when emissions are strongly correlated with RF, the constant lifetime scenario results in lower RF posteriors and thus smaller reductions in bank size relative to the time-dependent scenario. Because RF has high temporal correlation, the constant lifetime scenario used here has a consistently lower RF throughout. Prior to 1980, when the constant lifetime is lower than the SPARC

MMM, differences in production compensate for lower RFs, producing similar bank sizes between the two scenarios.

For CFC-12 (Fig. 1e), we see a smaller difference in bank size from the two lifetime scenarios, with the BPE bank again being much closer to the Ashford[9] and IPCC/TEAP(2005)[10] estimates than to the WMO(2003)[14] fixed-input top-down values of the late 1990s and early 2000s. Our values are again larger than Ashford[9] and IPCC/TEAP(2005)[10], and indicate a continuing bank of CFC-12 currently present. This contrasts with the most current WMO assessment's evaluation that the bank of CFC-12 has already been fully exhausted[3], although this conclusion is sensitive to the actual lifetime of CFC-12. While the SPARC MMM lifetime results in a higher bank estimate throughout the time period, the two CFC-12 BPE-derived bank estimates are within uncertainty of each other throughout the entire simulation period. This similarity in bank size occurs because the SPARC MMM lifetime has an averaged lifetime of 101.5 years over the period where observations are available (i.e. 1980–2016, see Supplementary Fig. 1), which is close to the constant lifetime estimate of 100 years for CFC-12. Similarly for CFC-113, the two lifetime scenarios do not result in significantly different BPE posterior bank estimates (see Fig. 1f). This is in part due to smaller time-dependent changes in lifetime (an average lifetime of 98 years from 1980 to 2016 for the SPARC MMM scenario versus a constant lifetime of 80 years, see Supplementary Fig. 1), but also due to larger relative uncertainties in modeled and observationally derived emissions for CFC-113 (i.e. larger σ × UB values relative to emissions). See Supplementary Fig. 9 for a comparison in the posterior distribution of uncertainties and relative uncertainties for each gas.

Figure 2 shows the reported production overlaid on top of the total calculated emissions for each of the chlorofluorocarbon gases considered here. This figure shows how emissions from the bank continue after global reported production becomes negligible (~2010), becoming the sole source of additional atmospheric emissions (unless unreported production is occurring). The figure underscores the importance of knowing how large the banks are in order to estimate whether or not observationally derived emissions exceed expectations following the Protocol, as well as future CFC concentrations and ozone recovery timescales. Recent studies have found that production of CFC-11 is likely continuing despite the Montreal Protocol phase-out[4]. Whether or not observationally derived emissions of other CFCs are consistent with expectations from the Protocol is also assessed below.

**Emissions estimates and discrepancies.** Figure 3 presents the observationally derived emissions (which depend upon the choice of lifetime, illustrated in the figure) along with posterior emissions from the Bayesian analysis (distributions of the residuals (i.e. $D_{\mathrm{emiss},t} - M(\theta_t)_{\mathrm{emiss}}$) are shown in Supplementary Fig. 10). The insets expand the results since 2010, when global production should have ceased under the Protocol. For CFC-11, under the reported production emissions scenario, observationally derived emissions are broadly consistent with the range of uncertainty in BPE banks from 2010 up to 2013. However, the simulated emission space does not encompass the increase in observationally derived emissions after 2012, consistent with findings in Montzka and colleagues[4]. When unexpected production is accounted for in prior production, the posterior emission space essentially encompasses observationally derived emissions (see Supplementary Fig. 11).

An important finding of Fig. 3 is that CFC-12 observationally derived emissions to date are broadly consistent with the analysis in this paper, suggesting that significant unexpected emissions are

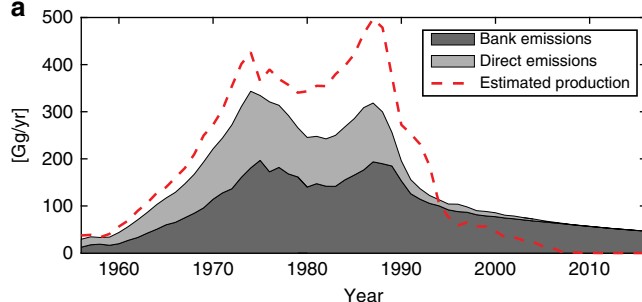

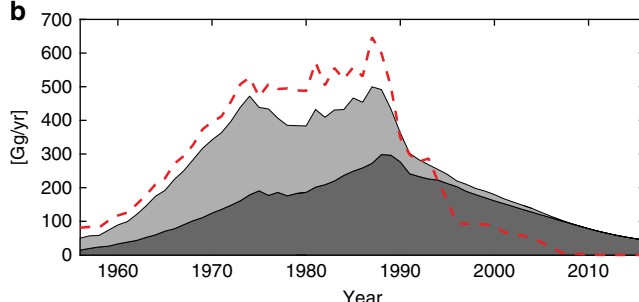

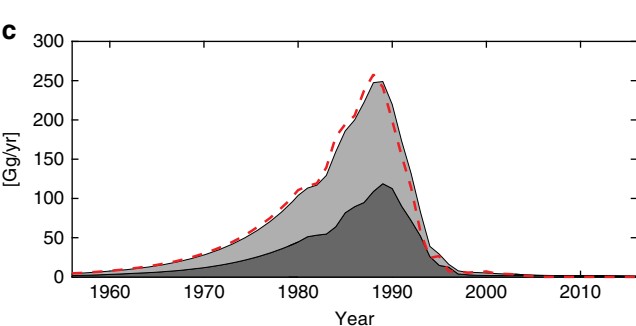

**Fig. 2 Reported production and estimated sources of emissions. a** Mean annual estimates of CFC-11 bank emissions (dark gray) and direct emissions (light gray) resulting from the BPE analysis using the SPARC multi-model mean lifetime assumption, and reported production to build the priors (i.e. we assume no large unexpected production post 2000). The red dashed line shows annual reported production values. (**b**) as in (**a**) but for CFC-12. (**c**) as in (**a**) but for CFC-113.

not needed to explain the behavior of that gas. It is interesting that the observationally derived emissions for both CFC-11 and CFC-12 lie at the lower edge of the Bayesian estimates from the mid 1990s to mid-2000s. Potential reasons for this joint behavior could include transient changes in circulation and hence lifetimes of both, or releases from stockpiles of both as phaseouts occurred, but other explanations such as larger errors in production are also possible. For CFC-113 on the other hand, there appears to be emission post-2010 that substantially exceeds this Bayesian analysis (discussed further below).

**Sensitivity of bank estimates to input parameters.** Note that the results of the BPE analysis are constrained by our choice of priors, which have been developed using published estimates of the input parameters. We investigate the sensitivity of our results to various input parameters. In particular we test the sensitivity of bank size to ~10% increases in the mean of the prior distributions of RF, DE for all equipment type in the bank (see Methods, Supplementary Methods 1, and Supplementary Tables 1–3 for details), as well a ~10% increase in the mean of the prior distribution of production. We also test the sensitivity of the bank to an increase

in the standard deviation of the RF prior distribution on closed cell foams, which are the largest component of the bank in recent decades. We find that BPE-derived bank estimates are moderately sensitive to production values and RF uncertainties. Production is not likely to be lower than the reported values, which were used to construct the base case scenario, and the lower bound of RF is fairly constrained, implying that our choice of priors are likely leading to conservative estimates in the size of banks (see Supplementary Fig. 12).

## Discussion

Understanding the trajectory of atmospheric CFC abundance in the coming years is key to understanding the timing of ozone hole recovery and future trends in radiative forcing of climate. While reported production of CFC-11 and CFC-12 has reduced to zero (or near zero), we can expect continued emissions from the current banks (Figs. 2 and 3). Accurate projections of atmospheric CFC abundances rely on knowledge of the quantity of banked CFC in existing equipment and products. Here we have provided a Bayesian uncertainty analysis of the bank size by integrating knowledge and uncertainties of CFC production quantities and equipment and product emissions functions, with observed concentrations of CFCs and atmospheric lifetime scenarios. Our analysis supports the view from bottom-up analyses that previous top-down estimates have underestimated CFC-11 and CFC-12 bank size (Fig. 1) by not accounting for uncertainties and likely biases in the parameters considered here (RF, DE, and Production), and not integrating all of these parameters into bank estimates. Another important finding is that substantial CFC-12 banks are likely still present, in contrast to recent WMO assessments[3] and current CFC-12 observationally derived emissions are broadly consistent with those expected from the banks according to our Bayesian model. Further, the emissions of CFC-11 are broadly consistent through 2012 but not beyond. This demonstrates that the constraints imposed on the CFC-11 priors from the current literature lead to posteriors that cannot feasibly reproduce the data. Since the model (and/or likelihood function) do not capture the full range of uncertainty, other factors must be at play. In particular, our analysis supports the finding that additional, unreported CFC-11 production after the 2010 global phase-out date mandated by the Montreal Protocol provides a more consistent emissions trajectory with observationally derived emissions, as suggested by Montzka and colleagues[4], but unexpected production is not required for consistency prior to 2012 when uncertainties are considered in detail. Further, our estimate of the unexpected total production associated with this emission would imply that the current CFC-11 bank size is approximately 140 Gg larger than it would otherwise be without the unexpected production assumption, implying ongoing additional contributions to ozone destruction in the future beyond those previously thought to be in the bank, even if further production ceases now.

Our study also underscores that emissions of CFC-113 significantly exceed expectations from banks alone after 2010 (see Supplementary Fig. 13 and Supplementary Note 1 for an analysis of uncertainties due to the lifetime of this gas). The absolute values of the total observationally derived emission averaged for 2005–2015 are relatively small for CFC-113, around 7.3 Gg yr$^{-1}$ (with a 1-sigma confidence interval ranging from 3.7 to 10.1 Gg yr$^{-1}$ using the uncertainty range in its lifetime estimated from the SPARC tracer-tracer correlation method). Further, uncertainties in the measured concentrations of this gas are larger than those of the other two. Nevertheless, it is notable that the emission of CFC-113 at about 7.3 Gg yr$^{-1}$ is comparable to the unexpected increase of emission of around 10 Gg yr$^{-1}$ for CFC-11 reported by Montzka and colleagues[4] after 2012. As noted earlier, CFC-113 is used as a

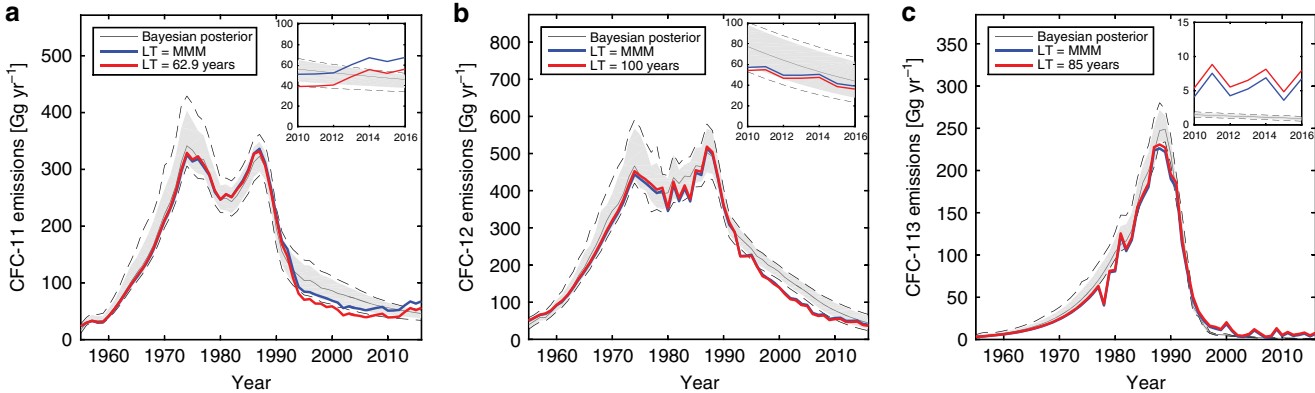

**Fig. 3 Observationally derived and posterior CFC emissions.** Emissions estimates are shown for (**a**) CFC-11, (**b**) CFC-12, and (**c**) CFC-113. In each panel, an inset shows results after 2010 while the main panels cover 1955 to 2016. Red and blue lines show results for observationally derived emissions using the SPARC MMM and constant lifetimes, respectively. The gray line indicates the mean Bayesian estimate, the gray shaded region indicates the 95% confidence interval and the dashed line indicates the 99% confidence interval.

feedstock for production of other chemicals, an allowed continuing use under the Montreal Protocol. According to the agreement, Parties are urged to keep feedstock leakage to a technically feasible minimum, which is thought to be of the order of 0.5%[15]. Global production of CFC-113 for feedstock use was reported to be about 131 Gg in 2014[15], implying emission of about 0.7 Gg yr$^{-1}$ at 0.5%, or about ten times less than our estimate. Figure 3 therefore suggests the need for further analysis of CFC-113 feedstock leakage as well as any potential for unreported non-feedstock production and use.

Given our BPE mean bank size estimates using the SPARC MMM lifetime, we next consider how different policy options would affect equivalent effective atmospheric chlorine (EESC) abundance for Antarctic chlorine, and future $CO_2$ equivalent emissions. Here we consider three different policy scenarios. Scenario 1 is a business as usual scenario. Under this scenario we assume a constant bank release fraction and bank size equal to that estimated in the last time period of the BPE simulation (i.e. the median RF in 2016 in the BPE simulation). We simulate emissions forward in time and estimate the chlorofluorocarbon abundance using the resulting emissions and the SPARC MMM atmospheric lifetime from 2010. As a test, we also consider 100% recovery and destruction of the CFC banks as an idealized best case. In Scenario 2 we consider an idealized upper limit in which there is 100% recovery and destruction of CFC banks in 2020 and no further emissions past 2020. Scenario 3 assumes that all banks are destroyed in 2000; this is an idealized "opportunity lost" emissions scenario where we consider CFC abundance with zero emissions following 2000. For each of the scenarios, we estimate the polar EESC following Newman and colleagues[1] with an average 5.5 year age of polar stratospheric air to account for the typical time required for air to reach the polar stratosphere from the surface. With the exception of CFCs, EESC values use mixing ratios from the WMO 2018 Assessment[3]. For CFCs, EESC values are estimated using mixing ratios from the WMO 2018 Assessment leading up to the scenarios. Results are shown in Fig. 4.

Figure 4a stacks contributions to EESC in a manner that optimizes understanding of what has dominated the recovery of EESC to date. The Figure makes clear that the bulk of the ozone recovery from the peak in EESC around 2000 to present is due to the global phaseout and rapid decline of $CH_3CCl_3$ (which has a global atmospheric lifetime of only about 5 years), along with substantial decreases in $CH_3Br$ and Halon concentrations. CFCs have declined slightly over this time, however, the contributions from CFC reductions can also be viewed as being offset to some extent by increases in EESC from the HCFCs that have replaced

them. Figure 4a illustrates that the fastest part of ozone recovery since peak depletion has already occurred. Future recovery is therefore increasingly dependent on reductions in CFCs, as well as other ODS reduction measures. Figure 4b stacks contributions differently to illustrate the gains in ozone recovery that could be obtained through recovery and destruction of CFC banks. These scenarios are all based on bank and emissions estimates using reported production (i.e. we do not include the unexpected emissions scenario). While Fig. 4a illustrates that CFCs have declined slightly from 2000 to present, the ongoing emission from banks (even without additional unexpected emissions) means that they have contributed less to the total reduction in EESC than they would have if the banks had been destroyed (e.g. Scenario 3 vs Scenario 2).

The year in which Antarctic EESC falls below 1980 levels is often used as a benchmark[3] to describe the path to ozone recovery, neglecting potential dynamical contributions. Using current estimates of lifetimes, polar EESC returns to pre-1980 levels in 2080 (scenario 1), 2074 (scenario 2), and 2067 (scenario 3). This comparison indicates that emissions from banked CFCs delay the recovery of the ozone hole by more than a decade compared to total destruction of the banks in 2000 and about seven years compared to current destruction. While 100% destruction of the banks is unrealistic, certainly some material can be recovered and destroyed (for example, via soil degradation of foams by careful burial in landfills instead of shredding[16]).

Our analysis demonstrates that CFC bank sizes are likely larger than what is currently assumed in the recent assessment[3]. Given the assumptions outlined above, we illustrate the effects of these larger bank sizes and the unexpected production scenario on projected mole fractions of CFC-11, 12, and 113 in Fig. 5 against the most recent projections. For CFC-11 in particular, the impacts on mole fraction projections can be substantial (e.g. a difference in 25 ppt), illustrating the importance of improved modeling of the banks for future international assessments. As a comparison, the WMO 2018 EESC projection results in Antarctic chlorine loading returning to 1980 levels by 2076. Our analysis indicates that scenario 1 projects a recovery by 2080, however, including the unexpected emissions scenario would result in a delay of an additional year (assuming that the source stops in 2019). An important assumption is that the unexpected emissions are only a fraction of the total production. Our analysis approximates that about 20% of total production makes up the unexpected emission, and the rest is initially banked. This would lead to long-term differences across scenarios with bank emissions as high as 49 Gg yr$^{-1}$ by 2030 if the unexpected production

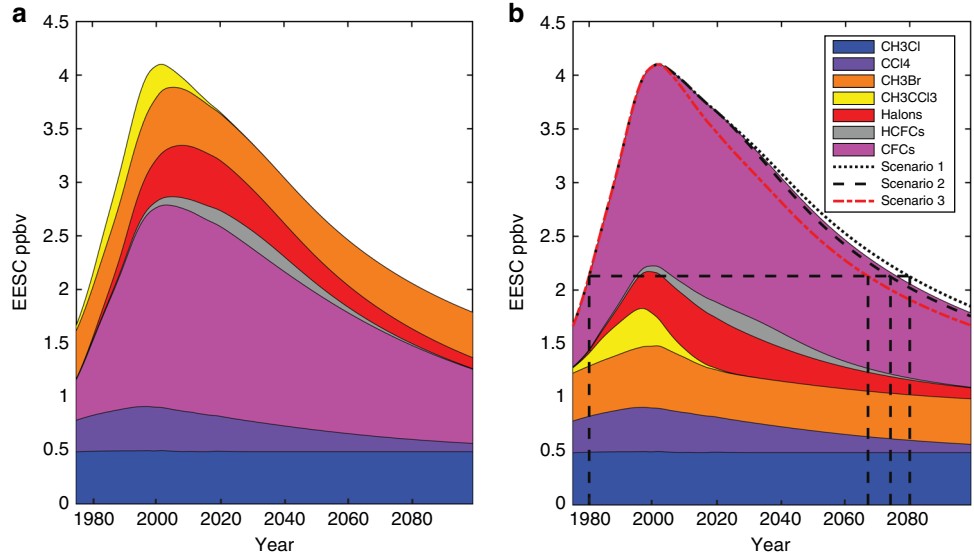

**Fig. 4 Measured and projected chlorine abundance and ozone recovery times.** Measured and projected Antarctic equivalent effective stratospheric chlorine (EESC) for all measured and projected abundances of ozone-depleting gases where mixing ratios come from the WMO 2018 assessment[3]. **a** EESC contributions are stacked in a manner that optimizes understanding of what has dominated the recovery of EESC to date. **b** EESC contributions are stacked with CFCs shown on top, including three scenarios for CFC−11, CFC-12, and CFC-113 constructed using mean bank emissions estimates resulting from the BPE analysis. Scenario 1 (dotted black line) represents the business as usual scenario, where bank emissions are simulated using the median release fraction (RF) and the median BPE estimated bank size in 2016. The RF is held constant over the entire simulation period. In scenario 2 (dashed black line) the banks are destroyed in 2020 with no further emissions. Scenario 3 (dashed red line) is the same as Scenario 2 except the banks are destroyed in 2000 followed by no further emissions. The SPARC MMM 2010 atmospheric lifetime is used to estimate the projected CFC abundance for each of the scenarios. EESC values leading up to the scenario simulations use mixing ratios from WMO (2018).

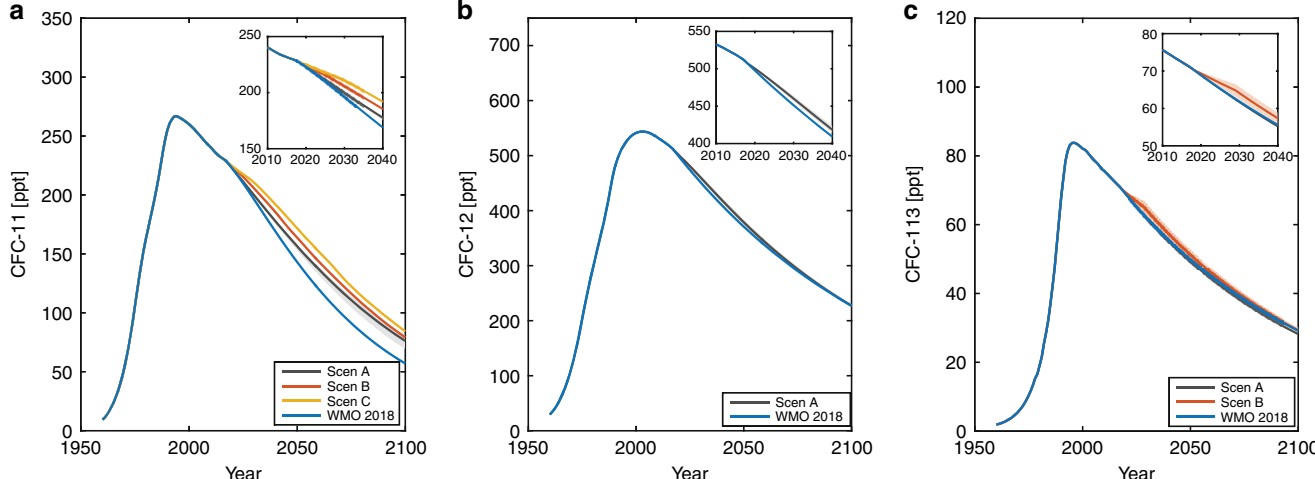

**Fig. 5 Measured and projected estimates of CFC concentrations.** Concentrations are shown for (**a**) CFC-11, (**b**) CFC-12 and **c** CFC-113. In each panel, an inset shows results from 2010 to 2040 while the main panels cover 1955 to 2100. For each panel, the blue line shows the WMO 2018 concentration estimates and projections. The black lines (Scen A in each panel) shows the concentrations projections using the median bank size and release fraction from our analysis starting in 2017 under the reported production scenario. The shaded gray region represents 1s.d. of uncertainty due to uncertainties in bank estimates. For (**a**) Scen B is equivalent to Scen A, except allows banks to account for the unexpected emissions scenario from 2000 to 2019, and Scen C is equivalent to Scen B except it allows the unexpected emissions to continue to 2029. For (**c**), Scen B allows for an additional 7.2 Gg yr$^{-1}$ of production until 2029 with the shaded region representing 1-s.d. of uncertainty in continued production (±5 Gg yr$^{-1}$).

continues unchecked for another decade, compared to about 32 Gg yr$^{-1}$ bank emissions for a scenario with no unexpected production.

Finally, we examine implications for global warming based upon carbon dioxide equivalents (CO$_2$eq) for a 100-year time horizon[17]. Table 1 shows the 21st century mean cumulative

emissions for three scenarios described above and corresponding mean cumulative CO$_2$eq emissions. The estimated future emissions from current banks could lead to an additional 9 billion metric tonnes of CO$_2$eq in global warming potential between 2020 and 2100, illustrating the importance of recovery and destruction of as large a fraction of the bank as is feasible and

**Table 1 Greenhouse gas contributions for example bank destruction options.**

| 2000-2100 cumulative emissions | Scenario 1: business as usual | Scenario 2: destroy banks in 2020 | Scenario 3: no banks after 2000 |
| --- | --- | --- | --- |
| CFC-11 | $2568 \times 10^3$ tonnes | $1245 \times 10^3$ tonnes | 0 |
| CFC-11* | $2911 \times 10^3$ tonnes | $1448 \times 10^3$ tonnes | 0 |
| CFC-12 | $2155 \times 10^3$ tonnes | $1864 \times 10^3$ tonnes | 0 |
| CFC-113 | $46 \times 10^3$ tonnes | $41 \times 10^3$ tonnes | 0 |
| CFC-11 $CO_2$eq | $1197 \times 10^7$ tonnes | $580 \times 10^7$ tonnes | 0 |
| CFC-11 CO2eq* | $1357 \times 10^7$ tonnes | $675 \times 10^7$ tonnes | 0 |
| CFC-12 CO2eq | $2198 \times 10^7$ tonnes | $1902 \times 10^7$ tonnes | 0 |
| CFC-113 CO2eq | $27 \times 10^7$ tonnes | $24 \times 10^7$ tonnes | 0 |
| Total CO2 eq | $3422 \times 10^7$ tonnes | $2505 \times 10^7$ tonnes | 0 |

Integrated emissions for bank destruction policy scenarios and their $CO_2$ equivalents for a 100-year time horizon (following values from https://www.ghgprotocol.org/sites/default/files/ghgp/Global-Warming-Potential-Values%20%28Feb%2016%202016%29_1.pdf) The CFC-11* case indicates estimated values for the unreported emissions scenario.

efficient. Avoiding the emission of CFC-113 of 7 Gg yr$^{-1}$ over the past decade (Supplementary Fig. 13, and Supplementary Note 1) would have represented about 0.4 billion tonnes $CO_2$eq. As illustrative example comparisons of upper limits of benefits, the European Union's cumulative projected greenhouse gas reductions under their Paris agreement pledge by 2030 relative to 2019 is ~7 billion metric tonnes[18] while the cumulative avoided emission of $CO_2$eq of HFCs from 2020 to 2050 under the Kigali amendment to the Montreal Protocol is ~53 billion metric tonnes (WMO, 2018)[3]. The opportunity lost already by not destroying the CFC banks in the year 2000 represents 25 billion metric tonnes of $CO_2$eq emissions since 2000 and delayed ozone hole recovery by an additional 7 years, illustrating the importance of prompt actions to the extent practical and efficient. Recovery and destruction of discarded or obsolete CFC banks benefits the climate system. However, we note that to optimize net gains for climate in systems that are still in use, a full life cycle analysis, taking account of factors including for example how existing foams contribute to energy efficiency, must be weighed against the $CO_2$eq content of the banks.

## Methods

**Background and motivation.** Top-down estimates of the banks are the cumulative sum of the difference between production and emissions since the onset of CFC production[7]. A noted challenge with the top-down approach is that it depends on small differences between large values (cumulative emissions and production) and requires both highly accurate reported production and observationally derived emissions for accurate results. Several studies suggest that uncertainties in production could be substantial, as discussed further below. Uncertainties in emissions depend on the accuracy of measurements of CFC abundances in the global atmosphere, and atmospheric lifetimes (discussed further below).

We can estimate annual global emissions, $D_{emiss,t}$, as

$$D_{emiss,t} = A\left([CFC]_{t+1} - [CFC]_t \times \exp\left(-\frac{\Delta t}{LT_t}\right)\right), \quad (1)$$

where $[CFC]_t$ refers to the concentrations of the particular CFC in year $t$, $LT_t$ is the atmospheric lifetime in year $t$, $\Delta t$ is equal to 1 year, and $A$ is a constant converting units of atmospheric concentrations to units of emissions. The time step is small enough that this is an accurate representation for the long-lived gases considered.

The bank estimates using the top-down approach can then be estimated as

$$Bank_t = \sum_{y=y1}^{t} (Prod_t - D_{emiss,y}), \quad (2)$$

where $y1$ is the first year of CFC production, and $Prod_t$ is the estimated production value in year $t$.

An alternative, bottom-up method could make use of information regarding a reference bank size starting point for a specific year as well as annual production, bank release fraction (i.e. the fraction of the existing bank that is emitted each year, composited across different applications), and direct emissions (the fraction emitted essentially immediately, in applications such as sprays, or through leakage). Using this approach, the bank size in year $t$ could be estimated recursively as

$$Bank_t = (1 - DE_t) \times Prod_t + (1 - RF_t) \times Bank_{t-1}, \quad (3)$$

where $RF_t$ is the bank release fraction, $Bank_{t-1}$ is the size of the bank in the previous year, $DE_t$ is direct emissions (i.e. the fraction of production in year t that is directly emitted that year, such as through leakage in the production process), and $Prod_t$ is the amount of CFC that is manufactured in year $t$. Estimates of chlorofluorocarbon bank size with this approach are therefore dependent on knowledge of production over time, the partitioning of production across different types of manufactured goods, as well as accurate assessments of the rate of release of ODSs for each type of manufactured product. Velders and Daniel[19] use the estimates of bank sizes from bottom-up inventory analysis as a starting point in 2008 using the findings of Ashford and colleagues[9] and IPCC/TEAP (2005)[10], and show how uncertainties in the different input parameters from Eq. (3) result in significant future uncertainties in bank size.

Here we adopt a Bayesian approach throughout the period considered. Our approach for discerning bank size may be thought of as a hybrid between the top-down and bottom-up that includes a wider range of constraints by making use of the information from both approaches and provides probabilistic outcomes. Using the input parameters from Eq. (3), we employ an alternative estimate to Eq. (1) by modeling emissions, $M(\theta_t)_{emiss}$, as

$$M(\theta_t)_{emiss} = RF_t \times Bank_{t-1} + DE_t \times Prod_t, \quad (4)$$

Where $\theta_t$ is the vector of input parameters ($RF_t$, $DE_t$, $Prod_t$, and $Bank_{t-1}$). With the exception of $Bank_{t-1}$, prior probability density functions for these parameters are constructed using a combination of probabilistic estimates of application-specific and time-dependent release fraction estimates from Ashford and colleagues[9], the distribution of production across equipment type from AFEAS (Alternative Fluorocarbons Environmental Acceptability Study) data (see for example https://unfccc.int/files/methods/other_methodological_issues/interactions_with_ozone_layer/application/pdf/cfc1100.pdf), and total production from the AFEAS (2001) and UNEP databases. The prior distributions for Bank input parameters are not independently defined. Instead they are simulated as a function of prior distributions for all previous timesteps of RF, DE, and production. They can be estimated by iterating Eq. (3) forward in time, or equivalently;

$$Bank_t = (1 - DE_t)Prod_t + \sum_{y=y1+1}^{t-1} (1 - DE_y)Prod_y \prod_{j=0}^{t-y-1} \left(1 - RF_{t-j}\right) + Bank_{y1} \prod_{j=y1+1}^{t} \left(1 - RF_j\right), \quad (5)$$

where $y1$ is the first year in the simulated time period.

Prior work has estimated banks using Eq. (2) either throughout the entire production record[14] or after 2008 using the inventory estimates of bank sizes for that year[3,19], but has not provided a statistical framework to constrain uncertainties in manufacturing parameters using uncertainties in CFC concentrations. Here we provide a probabilistic estimate of bank size by making use of Eq. (1) to constrain the distribution of Eqs. (3) and (4)'s parameter space in a Bayesian framework referred to as BPE. This allows us to assess whether the bottom-up and top-down approaches are consistent within estimated uncertainty, or if additional factors (e.g. fugitive emissions or stockpiled production) are necessary to reconcile the two approaches.

**Model framework.** To estimate the distribution of the parameters in Eq. (3), we use a form of Bayesian analysis called Bayesian melding that was designed by Poole and Rafferty[20] to apply inference to deterministic simulation models. It allows us to infer parameter estimates by taking advantage of the information available from both observed concentrations and the mechanistic simulation model of the bank, emissions, and concentrations comprised by (1), (3), and (4), hereafter termed simulation model. We employ a version of this method for input parameter uncertainty outlined in Bates and colleagues[21] and implemented in Hong and colleagues[22], which we henceforth refer to as Bayesian Parameter Estimation

(BPE). Because we are interested in the effects of atmospheric lifetimes on the range of bank outcomes, we implement the BPE algorithm separately for various assumed lifetimes. In the simulation model, we simulate bank size and emissions time series recursively, assuming an initial bank size in 1955 ($t=1$) equal to that estimated in WMO (2003)[14]. Bank sizes in 1955 are small enough that uncertainties in this number are insignificant. Bayesian updating is then implemented simultaneously for all time periods with available observations (1981–2016); therefore, the estimate for the bank in each year is based on all available observations.

We obtain posterior distributions for the vector of input parameters, $\theta$, by implementing Bayes' theorem as follows:

$$P\left(\theta|D_{\mathrm{emiss},1}, \ldots D_{\mathrm{emiss},N}\right) = \frac{P(\theta)P(D_{\mathrm{emiss},1}, \ldots D_{\mathrm{emiss},N}|\theta)}{P(D_{\mathrm{emiss},1}, \ldots D_{\mathrm{emiss},N})}, \quad (6)$$

where $P(\theta)$ describes the joint prior distribution of the input parameters (RF, DE, Production, and Bank) and $P\left(D_{\mathrm{emiss},1}, \ldots D_{\mathrm{emiss},N}|\theta\right)$ is the multivariate likelihood of all observed emissions given the input and output parameters of the simulation model. Each of the input parameters are $N\times1$ vectors, where $N$ is one less than the number of years of mole fraction observations used in the analysis (1980 to 2016). RF and DE are modeled jointly and assumed independent of Prod. Bank is modeled using Eq. (3) and therefore depends on all input parameters.

To solve Eq. (6), the general BPE model flow is implemented as follows. Begin by specifying prior distributions for input parameters. Next, using Monte Carlo simulation, sample from the prior distributions of the input parameters to simulate prior time series distributions for the simulation model outputs, emissions and bank size. We then specify the likelihood function of emissions from observed mole fraction and assumed lifetime. And finally, we estimate the posterior parameter distributions by implementing a sampling procedure. Each step of this model flow is described in more detail below.

**Atmospheric Lifetimes.** Assumptions about atmospheric lifetimes can have substantial impacts on CFC top-down estimates of bank size (see Fig. 1 for example). Many evaluations of CFC lifetimes[23,24] employed simple steady state models[23,24]. Understanding of atmospheric lifetimes has advanced through a recent assessment using three-dimensional models to better evaluate the time-dependent lags between tropospheric and stratospheric mixing ratios as emissions change (SPARC, 2013); that assessment showed that time dependent lifetime changes are substantial. To explore the impact of lifetimes and their time dependence on bank size we run the BPE using (i) constant lifetimes for each gas from the values in WMO 2003, (ii) time-dependent transient global lifetimes estimated by global photochemical models (taken from SPARC, 2013[12] which have mean values between 1960 and 2010 of 62.5, 113, 107 for CFC-11, -12 and -113, and (iii) constant lifetimes equal to the mean time-dependent lifetimes extended over the time period of the analysis (1955 to 2016). Values for WMO (2003) represent the last scientific assessment using the top-down approach without imposed constraints from bottom-up information provided by Ashford and colleagues[9] and IPCC/TEAP(2005)[10]. For purposes of comparison, we therefore adopt the values from WMO(2003)[14] of atmospheric lifetimes of 45 yrs, 100 yrs, and 85 yrs for CFC-11, CFC-12, and CFC-113, respectively. For the time-dependent lifetime scenario, we adopt the SPARC multi-model mean values, shown in Supplementary Fig. 1. Note that SPARC modeled lifetime estimates begin in 1960 and end between 1998 and 2010, depending on the model. Because we require a lifetime estimate for all years between 1955 and 2016, we extend each model's initial values from 1960 to earlier lifetimes (i.e. from 1955 to 1959), and extend their end values to all subsequent years until 2016. The time-dependent lifetime is then taken to be the mean of these extended modeled lifetimes.

**Priors for input parameters.** Implementing the BPE model requires a joint prior probability distribution to reflect our initial estimate of the uncertainty space of the input parameters, including production, direct emissions and bank release fraction based on the bottom-up methodology described above. We note that this approach, rather than developing uninformative priors, is intended to constrain the BPE results based on literature values. This allows us to assess the consistency of the top-down and bottom-up approaches within estimated uncertainty ranges. Time series of the prior and posterior distributions for each of the parameters are shown in the SM (Supplementary Figs. 14-16). We describe the choices of input parameter prior distributions below.

**Production.** Estimates for production typically rely on industry reported values (from the AFEAS database) or country level values (UNEP database), however, these estimates should be viewed with caution. Production from the former Soviet Union was not included in AFEAS and increases these values in earlier years by as much as about 20%[25]. In addition, by 2000 significant production in major developing countries was also not included in AFEAS. In broad terms, we expect reported values to underestimate true production values, as some of a growing number of producers may be omitted from national inventories, and some studies have probed possible black-market production of CFCs[26].

We build our prior distributions of global production based on reported values from AFEAS for years prior to 1989, and from UNEP from 1989 onwards. We adopt a correction for AFEAS data following WMO (2002)[14] (henceforth referred to as AFEAS/WMO), where AFEAS production values are augmented with production data from UNEP. Prior to 1989, companies reported their production of each molecule to AFEAS as part of the manufacturers' association. From 1989 onwards, countries reported national production values to the UNEP and were expected to meet the Protocol's reduction targets relative to 1986 values. Inconsistencies in accounting or reporting practices between different countries are possible, as are simple omissions depending upon the number of manufacturers and national regulatory mechanisms.

Given the potential biases discussed above, we construct our production priors under the assumption that these reported and adjusted annual production values are likely to be lower than the true total production in any given year. Our production prior follows a lognormal distribution such that:

$$\log(X_1, X_2, \ldots X_N) \sim N(\mu, \Sigma)$$
$$\mathrm{Prod}_t = B * \mathrm{Prod}_{0,t} * X_t + 0.95 * \mathrm{Prod}_{0,t} \quad (7)$$

where $N$ is the number of years considered in the model, $\mu$ is equal to zero for each year, and $\Sigma$ is a covariance matrix constructed with autocorrelation parameter ($\rho_1$) such that diagonal elements are equal to 0.25 and off-diagonal $d$ years apart are equal to $0.25 * \rho_1^d$. $\mathrm{Prod}_{0,t}$ is the reported production value in time period, $t$. $B$ is a constant that controls the uncertainty range which we set to 0.2 for production prior to 1989 when AFEAS/WMO data is adopted for reported production, and to 0.1 for production after 1989, when UNEP data is adopted. The higher uncertainty in the upper bound for the AFEAS/WMO data reflects our larger degree of uncertainty due to unreported production noted above, especially before the Protocol entered into force in 1989[27]. $\log(X_1, X_2, \ldots, X_N)$ are lognormally distributed random variables used to reflect our prior assumption that true production is not likely to be lower than reported and has a probability, albeit low, of being substantially higher than reported (e.g. for $B = 0.1$, there is a 3% probability of sampling above $1.2\times\mathrm{Prod}_{0,t}$). See Supplementary Fig. 17 for an illustration of the distribution.

Because we do not have data on the autocorrelation in the covariance matrix representing the uncertainty in reported production values, we estimate $\rho_1$ as an additional hyperparameter. Including this hyperparameter reflects our belief that there is some degree of consistency in underreporting across time. We assume the autocorrelation parameter PDF follows a Beta distribution (shown in Supplementary Fig. 18) as follows:

$$\rho_1 \sim 0.5 + 0.5 * \mathrm{Beta}(2, 2). \quad (8)$$

Note that the lower bound on the prior distribution for $\rho_1$ is greater than zero for computational efficiency; initial tests of the model found near-zero posterior probabilities for values lower than 0.5.

In light of recent work suggesting unexpected emission of CFC-11 after 2012[4], we also build an alternative production prior for CFC-11 to test how additional unreported production of this gas could impact bank size and emissions. For this unexpected emissions scenario, we assume an upper bound for added production based on the estimate from Montzka and colleagues[4] of unreported emissions after 2012 as high as 13,000 tonnes yr$^{-1}$. Based on our assumption of mean direct emissions (see below for details) of 21% of production for any year following 2000, this would equate to an upper bound of ~61,000 tonnes of CFC-11 produced in 2014 (i.e., 79% of production would be banked in that year). For this emissions scenario, we assume a linear increase in the upper bound of the unexpected production from 0 in 2000 to 61,000 tonnes by the end of 2012 and held constant thereafter. To reflect our adopted uncertainty in production, from 2000 onwards we assume a uniform distribution with a lower bound of reported production and an upper bound as described above. If the direct emission of this unexpected production were higher, or if the production were higher for applications that released CFC-11 quickly, this total production figure would be smaller, perhaps substantially so.

**Direct emissions and bank release fraction.** We estimate annual direct emissions and the bank release fractions jointly using a bottom-up accounting of the various equipment types comprising the bank, their relative prevalence, and the unique loss rates at which they emit CFCs. DE and RF are assumed to be stationary and unique for each equipment type (e.g., open cell foams, closed cell foams, chillers, etc.,) with their respective uncertainties and loss functions as shown in the supplement (Supplementary Tables 1-3). RF for the total bank is time dependent as it depends on the composition of the bank, which changes over time. DE is modeled as a fraction of total production in a given year. Therefore, DE in year $t$ depends on how production is apportioned across equipment type in year $t$ combined with the loss rate for each equipment type in its first year of life. RF is modeled as the fraction of the bank that is released in a given year and therefore depends on the composition of the bank. Thus, RF depends on the relative prevalence of each equipment type, and their unique loss rates in all prior years.

To estimate DE and RF, we first develop priors for annual production and unique loss rates for each equipment type. The priors for production for each equipment type are developed using production data from AFEAS,(2001)[28] which provides data on both total reported production of each CFC molecule in

any given year from 1930 to 2000, as well as the breakdown of total production into the types of applications. After AFEAS data ends in 2000, we use that year for the priors in each subsequent year. The one exception is when constructing CFC-11's unexpected emissions scenario. Because we have no knowledge about the applications for which this new production is being used, our prior assumption is that each equipment type is equally probable following 2000. For the loss rate parameters, we use chlorofluorocarbon release rates from Ashford and colleagues[9] to construct priors. For each type of equipment, Ashford and colleagues[9] construct estimates of loss rates over time by type of product for each molecule. For example, they estimate that 50% of CFC-11 used in aerosols and solvents are emitted the year they are produced, and 50% is emitted the following year. In contrast they estimate closed cell foam releases at 3.66% of its bank each year. For more details on these priors, and how the RF and DE sample time series are constructed refer to the Supplementary Methods 1 and Supplementary Tables 1-3.

Note that for the unexpected emission scenario, the assumption of equally probable production across equipment types leads to a wider and time-varying range of RF and DE sampled values than for all other scenarios. The result of jointly constructing RF and DE time series in this manner is that both parameters are constructed to exhibit covariance and temporal correlation for physical consistency. Also note that for total production, we use AFEAS data up until 1989, after which we use UNEP data. For estimating RF and DE, production data from AFEAS is used only to approximate relative production by equipment type over time. This, in turn, provides a prior estimate of the relative distribution of equipment type in the bank, which we use to estimate RF and DE. These RF and DE priors are constructed independently of total production priors.

**Specifying the Likelihood function**. For each atmospheric lifetime scenario, emissions are inferred from observed global mole fractions using Eq. (1). We henceforth refer to these as observationally derived emissions, or data, $D_{\text{emiss},t}$, where $t$ refers to the year. Observations come from the merged AGAGE and NOAA global surface mean mole fraction in ref. [29] and are available from 1980 to 2018. We assume that:

$$D_{\text{emiss},t} = M(\theta_t)_{\text{emiss}} + \sigma_t, \qquad (9)$$

where $M(\theta_t)emiss$ is the modeled emissions following Eq. (4), $\theta_t$ is the vector of input parameters (RF$_t$, DE$_t$, Prod$_t$, and Bank$_{t-1}$), and $\sigma_t$ is the error term assumed normal with mean zero and covariance $S_t^2$. The likelihood function is therefore a multivariate function of the difference between modeled and observationally derived emissions:

$$P(D_{\text{emiss},1}, \cdots, D_{\text{emiss},N}|\theta) = \frac{1}{(2\pi)^{\frac{N}{2}}\sqrt{|S|}}\exp\left\{-\frac{1}{2}\Delta^T S^{-1}\Delta\right\}, \qquad (10)$$

where $\Delta$ is an NxN diagonal matrix with diagonal elements;

$$\Delta_{t,t} = D_{\text{emiss},t} - M(\theta_t)_{\text{emiss}}, \qquad (11)$$

$S$ is a covariance matrix representing the sum of the observationally derived and modeled emissions uncertainties. While there exist published estimates of observationally derived uncertainties[30] we have no prior information on modeled uncertainties. We therefore estimate S as follows: All diagonal elements are equal to $\sigma \times$ UB where UB is set equal to the larger value of 40 Gg yr$^{-1}$ or twice the mean difference in emissions inferred from observations using the maximum time-varying SPARC lifetimes and minimum time-varying SPARC lifetimes. $\sigma$ is a parameter that is estimated from a Beta prior distribution with input parameters $\alpha = \beta = 5$. The prior and posterior distributions for $\sigma \times$ UB are shown in Supplementary Fig. 9. Off-diagonals of $S$ are estimated using an autocorrelation hyperparameter, $\rho_{\text{err}}$, drawn from a Beta distribution with parameters $\alpha = \beta = 2$ with a lower bound of 0.5 and upper bound of 1.

**Estimating posteriors**. Because the analytical form of the posterior is intractable, we use the sampling importance resampling (SIR) method to approximately sample from the marginal posterior distributions[21,22,31]. This method involves sampling from the prior and then resampling the prior samples according to an importance ratio. For a detailed description of SIR, refer to the work of Hong and colleagues[22]. We implement the SIR method by drawing 1,000,000 samples from the prior and then resampling from these samples 100,000 times to obtain the posterior distribution. These sample sizes were chosen such that multiple sampling estimates produced consistent results for prior and posterior bank distributions.

## Data availability
The datasets generated and/or analyzed during the current study are available at https://github.com/meglickley/CFCbanks.

## Code availability
All code used in this work is available at https://github.com/meglickley/CFCbanks. All analyses were done in MATLAB.

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

## Acknowledgements

M.J.L. and S.S. gratefully acknowledge support by a grant from VoLo foundation.

## Author contributions

M.J.L., S.S., G.J.M.V. and J.D. conceptualized the work. M.J.L., S.S., S.F., and M.R. designed the work. M.J.L. conducted the analysis. G.J.M.V., J.D., M.R. and K.S. acquired the data. All authors contributed to the interpretation of the data. M.L., S.S. and S.F. drafted the manuscript. All authors contributed substantial revisions of the manuscript.

## Competing interests

The authors declare no competing interests.
