## [Peer Review File · Nature Communications]

Reviewers' comments:

Reviewer #1 (Remarks to the Author):

The choice of a Bayesian model is logical in view of the balance that needs to be struck between measured data and an understanding of the processes that lead to banking of ODS and the subsequent emissions from those banks. The adoption of a Bayesian model is also a novel approach in the context of the determination of ODS bank sizes and resulting emissions.

It is gratifying to see that the conclusions of the current work support the validity of the bottom-up bank and emissions estimates made in the 2003-2005 period by TEAP and IPCC and that concerns over the discrepancies between top-down and bottom-up estimates at the time have now been reconciled. It is notable that TEAP made efforts to establish some explanation for this discrepancy as far back as October 2006 in the Task Force Report on Emissions Discrepancies and it might be appropriate to make some reference to this report in recognition of that earlier work, which highlighted the sensitivity to atmospheric lifetime of the top-down models of the time and postulated that an increase from 45 years to 65 years would be sufficient to reconcile the discrepancies. This is close to the 62.9 years now apparent from the average SPARC MMM lifetime. At the moment, I don't see the Task Force on Emissions Discrepancies Report referenced in the discussion or in the references unless I have missed it somewhere.

The other significant comment I would make is that the bank scenarios set out in the Discussion and Conclusions section don't take account of the distinction between those elements of the bank that remain in products continuing to bring societal value and those which have already reached the waste stream. Among the former are banks in insulating foams which are continuing to contribute efficiently to energy savings in numerous buildings around the world. The idea of stripping those out prematurely to minimize bank sizes going forward needs to be balanced against the on-going climate benefit of them staying in-situ until their natural lifecycle reached. There are risks of significant accelerated emissions resulting from trying to removed installed foams prematurely.

Paul Ashford

Reviewer #2 (Remarks to the Author):

Review of Quantifying contributions of chlorofluorocarbon banks to emissions and impacts on the ozone layer and climate by Lickley et al.

The paper presents a new approach, Bayesian Probability Estimation (BPE), to estimating the atmospheric emissions and banks of CFCs which combines top-down and bottom-up approaches.

This enables a more robust quantification of emissions and their uncertainties than previously reported. It leads to the following conclusions: the banks of CFCs are larger than previously thought meaning that there is greater potential for further emissions, delaying the recovery of stratospheric ozone and contributing more to radiative forcing; global emissions of CFC-11 are consistent with release from banks prior to 2012, but there are unexpected emissions after then; emissions of CFC-12 are consistent with the banks; emissions of CFC-113 from banks can only explain part of the emissions implied by top-down estimates. This paper is a very important contribution to the current scientific debate on CFC. The results are of interest to scientists in the field as well as international policy makers and so I recommend publication once the issues below have been addressed.

Something that I struggle to understand is why the BPE method leads to both larger banks (Fig. 1) and what seems to be larger emissions (Fig. 3) of CFC-11 and CFC-12 than derived using the top-down approach. My first thought was that if the banks are larger than previously thought, then emissions to date may have been smaller, but this seems not to be the case. This seems to imply that different production rates were used for the two methods (the way the production rates are calculated for the BPE are described in the Methods and Supplementary Material, but not what production rates were used for the top-down calculations) and importantly that the production in the BPE is larger than for the top-down estimates. If one of the reasons for the banks being larger than previously thought is because the production is calculated to be larger, then this seems to me to be an important message that should be made. If this is not the reason, then some other explanation as to why both the banks and emissions are larger with the BPE is required.

The point about the production is important as it has implications for how one might interpret the implications of the larger banks. In theory, if more of the CFCs are still in the banks, then we have a greater opportunity to recover and destroy the CFCs before they are released to the atmosphere and in so doing limit damage to ozone and climate more than was thought possible. If the larger banks are due to past underestimated production, then the potential for damage is greater than previously thought.

The results are appropriately discussed in the context of previous literature.

In general, the manuscript is well written, but there are several places where the manuscript could be improved for clarity and for consistency in presentation. Overall the methodology is mostly clear, although it was not always clear in the main text of the paper and it was not until reading the Methods and Supplementary Material later on that some things were clarified. Suggestions of how to improve this are below.

Lines 29-30. The statement about the impact of the CFCs should be qualified with the addition of "if the CFCs in these banks are not entirely recovered".

Lines 46-48. I think it would be helpful to the reader to introduce the concepts of banks here and to explain that continued emissions of CFCs from banks are expected long after production has ceased. This along with their long lifetimes is why the chlorine loading from these chemicals is declining slowly.

Line 58. "causes of measured".

Line 71. Delete ",".

Lines 83-84. Is there a difference between estimates of production (line 78) and of sales? If there is this, then this can explain some of the difference between the top-down and bottom-up estimates.

Lines 119-120. Suggest adding “if not entirely recovered” to the sentence about climate change.

Lines 127-128. The SPARC MMM runs only went up to 2010 (Fig. S1) and from what I can make out from the SPARC report the MMM was only calculated up to 2006 as 2 models did not run beyond that date (Fig. 5.13 in the SPARC report). How was the SPARC MMM extended to 2016 and prior 1960? It would be helpful to point the reader to Fig. S1 here.

Line 144. DE is first defined here simply as direct emissions (which suggest units something like Gg/y), but it is not until line 449 that it is defined as a fraction. It would be helpful to clarify this earlier and possibly call it a direct emissions factor to differentiate it from absolute direct emissions.

Lines 169-170. The posteriors (Fig. S5) look very different to me, especially for 1985.

Lines 183-184. What impact does the time dependency of the SPARC MMM lifetime have on this? Might the divergence after 1990 be driven by the fact that the SPARC MMM time dependent lifetime reduces over the period 1960 to 2006 such that it exceeds the mean value of 62.9 years in the earlier years and in the later years is less than 62.9 years (Fig. S1). Hence, after 1990 when emissions are strongly related to release from the banks, the SPARC MMM run with a lifetime of <62.9 years at that time will lead to greater emissions and so leading to a smaller bank than a run with a fixed lifetime of 62.9 years. The different posterior release fractions would then be a reflection of this.

Line 198. It would be informative to state the average value of the SPARC MMM lifetime and to refer to Fig. S1.

Line 201. It would again be informative to state the average value of the SPARC MMM lifetime and to refer to Fig. S1.

Line 203. Fig. S6 shows $\sigma \times UB$ (emissions (Gg)). The values for CFC-113 are similar or slightly less than for the other gases. However, because the emissions of CFC-113 are less, this implies that its values of σ must be larger than for the other CFCs. Presenting a figure of σ only might illustrate this point better.

Line 207-221. The Fig. 1 caption needs some improvement. Please state what the Ashford, TEAP, WMO and IPCC/TEAP markers and lines represent, noting that they are bottom-up as this is how they are referred to in the text. The titles given to the plots whilst broadly correct could be improved for consistency of plots. Bank estimates or bank size. Plot (c) isn't just BPE. I suggest replacing “the production prior includes an unexpected emissions scenario” (Line 216-217) with “the production prior includes additional production to account for unexpected emissions”. The red line is not the only difference (Line 218) as (d) has the red dashed and blue dashed lines from (a) instead of the black line in (c). There is also no line for a run with unexpected emissions. Unlike (e) (Line 220) red dashed and blue dashed lines are not in (c). The unexpected emissions scenarios (Line 220) needs to be mentioned for (d) as well. Also, it would be good to mention the lifetimes used for CFC-12 and CFC113, both the fixed one and the average value of SPARC MMM LT noting that it is time varying.

Lines 253-255. The Bayesian analysis seems to suggest that emissions of CFC-12 were larger than the observationally-derived emissions from around the mid-1990s to ~2006. What are the implications of this? Interestingly, during this same period, the observationally-derived emissions for CFC-11 are at the lower boundary of the Bayesian confidence interval (or even below for the fixed lifetime). Could this be related to what is seen for CFC-12?

Line 260. Fig. 3. I think it would be informative to add a line for the scenario which includes unexpected emissions in the inset in (a).

Line 295. "implying ongoing contributions to ozone destruction in the future even if further production ceases now." This could be expanded to clarify that this ongoing contribution is in addition to the on-going contribution that we already expected from what had previously been thought to be in the bank.

Line 306. Full stop before "Global".

Line 313. Subscript for CO₂.

Line 316. Fig. 4 states "median" as opposed to "mean". Which is it?

Lines 319-321. It was not immediately clear to me that Scenario 2 is the same as the idealized best case. Suggest "In Scenario 2 we consider an idealized best case in which there is 100% recovery and destruction of the CFC banks in 2020 and no further emissions past 2020."

Lines 327-328. It would be helpful to clarify in the text that the source of the EESC data in Fig 4 (left (a)) is from the WMO 2018 Assessment. Lines 364-365 state that EESC values leading up to the scenarios are from the WMO, but am I correct in that the mixing ratios for all chemical species other than the CFCs come WMO, right up to 2100? Please clarify.

Lines 332-334. This comment makes a comparison between mixing ratios 2000 to present with or without banks being destroyed. Surely this is the comparison between Scenarios 1 and 2 which are illustrated in Fig 4 (right (b)). I would therefore suggest moving this comment to after where Fig. 4 (right) is described in the text.

Lines 355-365. Fig. 4. It would be helpful to have tickmarks on the x-axis. In the text the plots are referred to as left and right, but in the caption as (a) and (b). Please be consistent. I appreciate that there is a superscript pointing to the WMO 2018 Assessment for plot (a), but it would be helpful at this point to state the source as done in line 365. Is it the mean or median RF and bank sizes? The text refers to the mean RF.

Lines 374-378. This point about the difference between WMO 2018 EESC and the scenario 1 EESC is illustrated in Fig. 4. I suggest it would have been better to have made this point earlier when discussing Fig. 4. This would then separate it from the new point being made here which is that the unexpected emissions presented in Fig. 5 as Scen 2 extend the recovery by 1 year, from 2084 to 2085. It would help to refer to the scenario number in the text (see also next comment below). Please also clarify if the EESC plotted in Fig. 4 is "Antarctic" Cl loading as referred to here.

Lines 386-396. Fig. 5. Please explain in the caption what is represented by the grey shading around the black lines. I couldn't find mention of Scen 3 in the text, nor could I find mention of Scen 2 as it relates to CFC-113 (plot (c)) in the text. It was not easy to follow the different scenarios that were performed since those presented in Fig. 4 are called Scenario 1, 2 and 3, whilst those in Fig. 5 are called Scen 1, 2 and 3, and it appears that some of the scenarios with the same number are different between the figures and even between plots in the same figure. It would really help to have a set of unique scenario numbers and to ensure that all scenarios are discussed in the text, not just the figure captions.

Lines 421-422. What is the source of the production data used in the top-down estimates presented in this paper?

Line 489. "to apply"?

Line 536-538. How were the SPARC time dependent lifetimes extended to 1955 and 2016?

In the equations used in the Methods section and the Supplementary Material, I would recommend using the same names for production, either Production or Prod, unless these two terms are meant to represent something different, in which case please clarify.

Equation 3. Whilst this equation is correct wouldn't a more general equation to describe the bottom-up approach be to calculate the integral from in year y_1 to t , similar to equation 2?

In several places throughout the methods it is stated that $Bank_{t-1}$ is one of the input parameters. I struggled to understand how a prior can be used for $Bank_{t-1}$ as it would be time varying, dependent on how much has been released from the bank $((1-RF_t)*Bank_{t-1})$ and how much new production has gone into the bank $((1-DE_t)*Production_t)$. This was compounded by the description of the Priors for Input Parameters only describing production, RF and DE. Also, in Supplementary Material plots of the priors are only given for the production, RF and DE (Figs. S8, S9 and S10) and the headings only refer to Direct Emissions, Release Fractions and Production and not banks. I can see from lines 637-638 that the RFs depend on the composition of the bank so imply priors for the bank. Further description of the bank priors would be useful.

Supplementary Material

It is not clear to me why $EF_{k,y1}$ and $EF_{k,bank}$ are used on page 2 rather than $DE_{k,y1}$ and $RF_{k,bank}$.

Figs. S2, S3, S5. Presumably for CFC-11? Suggest using the same ranges for the scales of the axes to help comparison. What is the constant lifetime value, 45 years? Why are the priors different for the two different lifetime scenarios?

Fig. S4. Presumably for CFC-11? Units of production? Which BPE run (e.g. lifetime)?

Reviewer #3 (Remarks to the Author):

What are the major claims of the paper?

This paper utilizes Bayesian parameter estimation (BPE) to assess the uncertainty in CFC-11, CFC-12 and CFC-113 banks and bank emissions. The underlying models and distributions were built using expert knowledge available to the authors from a variety of sources, and the authors acknowledged and dealt with known and potential biases and inconsistencies in these sources. Interpretation of the posterior distributions and comparison of the posterior distributions to existing non-BPE approaches as well as observed emissions data, provided conclusions regarding potential delays to ozone hole recovery and questions regarding sources of CFCs. The authors illustrate how BPE can be used to provide better measures of the uncertainty in conclusions than can be provided by non-probabilistic approaches.

Are they novel and will they be of interest to others in the community and the wider field?

The application of BPE to CFC emissions is to my knowledge novel and extends the use of BPE to a new area. The authors make the case that the use of BPE in modeling emissions and banks better characterizes the uncertainties we have and could lead to different decision make processes in the real-world.

* Is the work convincing, and if not, what further evidence would be required to strengthen the conclusions? Comment on the appropriateness and validity of any statistical analysis.

Did the authors examine the distribution of the joint prior (or model induced outputs using jargon of Bates et al (2003) and Poole and Raftery (2000)) distribution of RF and Banks for the years 1995 and 2005? The joint RF and Banks posterior distributions for these years shown in Figure S4, would appear to favor a narrow ridge of values. Why do we think this is? The authors describe the correlation of RF and DE and both are involved in the model for Banks, however it is not obvious that the posterior should be this restricted. Figures S2 and S3 shows that the posterior distributions for RF and DE in 2005 consisted of values resampled from the lower end of prior distributions for these parameters in that year. (This effect is stronger for RF than DE.) Is this seen in the joint prior also? I.e. did the joint posterior consist of resampled values along the boundary of the prior distribution? If so, did the authors examine sensitivity of the final interpretations to widening this joint prior?

Did the authors examine sensitivity of the results discussed in the paper to minor perturbations in the parameters of the prior and hyperprior distributions? E.g. if such parameters were adjusted by 10%, what were the resulting changes in the posterior results? This is particularly important for parameters that are not well known.

* On a more subjective note, do you feel that the paper will influence thinking in the field?

I would like to hope that this paper makes a convincing enough argument to influence thinking in the field, specifically the questions it raises about the sources of CFC-113 and potential changes to the management of other CFC banks. BPE is, as the authors point out, useful in incorporating information from observed data as well as expert knowledge of the systems (through priors).

* Ability of a researcher to reproduce the work, given the level of detail provided.

The manuscript is written clearly with sufficient detail to encourage replication of the results. Supplementary materials are useful and relevant to the discussions. Replication would require access to any dataset that was not specifically enumerated in the materials.

Congratulations on a nice application of the method to a novel application. I particularly appreciated the grounding of conclusions and decisions in expert knowledge of the observed phenomena.

Addressing the questions raised above regarding sensitivity analysis and possible effect of the prior distribution choices, would strengthen the paper and apprehension that some might have to the use of this BPE approach.

In the following we include our inline responses to each comment in blue along with line numbers that refer to the lines in the track-changes version of the revised Manuscript.

Reviewers' comments:

Reviewer #1 (Remarks to the Author):

The choice of a Bayesian model is logical in view of the balance that needs to be struck between measured data and an understanding of the processes that lead to banking of ODS and the subsequent emissions from those banks. The adoption of a Bayesian model is also a novel approach in the context of the determination of ODS bank sizes and resulting emissions.

It is gratifying to see that the conclusions of the current work support the validity of the bottom-up bank and emissions estimates made in the 2003-2005 period by TEAP and IPCC and that concerns over the discrepancies between top-down and bottom-up estimates at the time have now been reconciled. It is notable that TEAP made efforts to establish some explanation for this discrepancy as far back as October 2006 in the Task Force Report on Emissions Discrepancies and it might be appropriate to make some reference to this report in recognition of that earlier work, which highlighted the sensitivity to atmospheric lifetime of the top-down models of the time and postulated that an increase from 45 years to 65 years would be sufficient to reconcile the discrepancies. This is close to the 62.9 years now apparent from the average SPARC MMM lifetime. At the moment, I don't see the Task Force on Emissions Discrepancies Report referenced in the discussion or in the references unless I have missed it somewhere.

We agree. We added this point at line 116-118:

A subsequent TEAP (2006) assessment¹¹ suggested that some of the discrepancy could stem from longer lifetimes than thought, a result supported by later stratospheric modeling analysis¹².

The other significant comment I would make is that the bank scenarios set out in the Discussion and Conclusions section don't take account of the distinction between those elements of the bank that remain in products continuing to bring societal value and those which have already reached the waste stream. Among the former are banks in insulating foams which are continuing to contribute efficiently to energy savings in numerous buildings around the world. The idea of stripping those out prematurely to minimize bank sizes going forward needs to be balanced against the on-going climate benefit of them staying in-situ until their natural lifecycle reached. There are risks of significant accelerated emissions resulting from trying to removed installed foams prematurely.

We agree. We modified text to make clearer that 100% is an idealized upper limit. We also modified the closing paragraph per this comment as follows (Line 734-746):

As illustrative example comparisons of upper limits of benefits, the European Union's cumulative projected greenhouse gas reductions under their Paris agreement pledge by 2030 relative to 2019 is ~ 7 billion metric tonnes¹⁶ while the cumulative avoided emission of CO₂eq of HFCs from 2020 to 2050 under the Kigali amendment to the Montreal Protocol is ~ 53 billion metric tonnes (WMO, 2016). The opportunity lost already by not destroying the CFC banks in the year 2000 represents 25 billion metric tonnes of CO₂eq emissions since 2000 and delayed ozone hole recovery by an additional 7 years, illustrating the importance of prompt actions to the extent practical and efficient. Recovery and destruction of discarded or obsolete CFC banks benefits the climate system. However, we note that to optimize net gains for climate in systems that are still in use, a full life cycle analysis, taking account of factors including for example how existing foams contribute to energy efficiency, must be weighed against the CO₂eq content of the banks.

Paul Ashford

Reviewer #2 (Remarks to the Author):

Review of Quantifying contributions of chlorofluorocarbon banks to emissions and impacts on the ozone layer and climate by Lickley et al.

The paper presents a new approach, Bayesian Probability Estimation (BPE), to estimating the atmospheric emissions and banks of CFCs which combines top-down and bottom-up approaches. This enables a more robust quantification of emissions and their uncertainties than previously reported. It leads to the following conclusions: the banks of CFCs are larger than previously thought meaning that there is greater potential for further emissions, delaying the recovery of stratospheric ozone and contributing more to radiative forcing; global emissions of CFC-11 are consistent with release from banks prior to 2012, but there are unexpected emissions after then; emissions of CFC-12 are consistent with the banks; emissions of CFC-113 from banks can only explain part of the emissions implied by top-down estimates. This paper is a very important contribution to the current scientific debate on CFC. The results are of interest to scientists in the field as well as international policy makers and so I recommend publication once the issues below have been addressed.

Something that I struggle to understand is why the BPE method leads to both larger banks (Fig. 1) and what seems to be larger emissions (Fig. 3) of CFC-11 and CFC-12 than derived using the top-down approach. My first thought was that if the banks are larger than previously thought, then emissions to date may have been smaller, but this seems not to be the case. This seems to imply that different production rates were used for the two methods (the way the production rates are calculated for the BPE are described in the Methods and Supplementary Material, but not what production rates

were used for the top-down calculations) and importantly that the production in the BPE is larger than for the top-down estimates. If one of the reasons for the banks being larger than previously thought is because the production is calculated to be larger, then this seems to me to be an important message that should be made. If this is not the reason, then some other explanation as to why both the banks and emissions are larger with the BPE is required. The point about the production is important as it has implications for how one might interpret the implications of the larger banks. In theory, if more of the CFCs are still in the banks, then we have a greater opportunity to recover and destroy the CFCs before they are released to the atmosphere and in so doing limit damage to ozone and climate more than was thought possible. If the larger banks are due to past underestimated production, then the potential for damage is greater than previously thought.

We agree that this is an important point that needs clarification. Our analysis indicates that banks and emissions are both larger than previously thought, in large part because production has likely been on average 13 % higher than reported. One source of additional production could be the former Soviet Union production not included in AFEAS that had been estimated to be as high as 20%, along with unreported developing country production and possible black market production (noted on lines 653-656 in the original draft). In response to this comment, we now note the influence of unreported production more explicitly in the text in Lines 267-270:

Importantly, our posterior estimates of production indicate that total production from 1955 to 2016 of CFC-11 has likely been 13% (1-sigma \cong 3%) larger than the values used in previous scientific assessments, contributing further to the discrepancies between the BPE bank size and WMO (2003) bank estimates.

The results are appropriately discussed in the context of previous literature. In general, the manuscript is well written, but there are several places where the manuscript could be improved for clarity and for consistency in presentation. Overall the methodology is mostly clear, although it was not always clear in the main text of the paper and it was not until reading the Methods and Supplementary Material later on that some things were clarified. Suggestions of how to improve this are below.

Lines 29-30. The statement about the impact of the CFCs should be qualified with the addition of "if the CFCs in these banks are not entirely recovered".

We added the wording 'left unrecovered' to the abstract to convey this point (Line 29)

Lines 46-48. I think it would be helpful to the reader to introduce the concepts of banks here and to explain that continued emissions of CFCs from banks are expected long after production has ceased. This along with their long lifetimes is why the chlorine loading from these chemicals is declining slowly.

We added clarifying language (Lines 52-55):

Further, CFCs were produced for use in equipment, some of which have lifetimes of up to multiple decades. This implies that a bank of material could still exist, contributing to current and future emissions.

Line 58. “causes of measured”.

This has been corrected (Line 64).

Line 71. Delete “,”.

This has been corrected (Line 98).

Lines 83-84. Is there a difference between estimates of production (line 78) and of sales? If there is this, then this can explain some of the difference between the top-down and bottom-up estimates.

We agree that this is a possibility, but do not have a basis for including it. It has been noted in the text at lines 170-171:

Differences between annual production and sales (e.g., stockpiling) are possible but are not included here due to lack of quantitative information.

Lines 119-120. Suggest adding “if not entirely recovered” to the sentence about climate change.

We changed the text to clarify that it must also be efficiently recovered due to potential GHG emissions from the recovery process: (Line 183-184).

if they are not efficiently recovered?

Lines 127-128. The SPARC MMM runs only went up to 2010 (Fig. S1) and from what I can make out from the SPARC report the MMM was only calculated up to 2006 as 2 models did not run beyond that date (Fig. 5.13 in the SPARC report). How was the SPARC MMM extended to 2016 and prior 1960? It would be helpful to point the reader to Fig. S1 here.

Thank you for pointing this out, it is true that we didn’t explain how we extended the modeled values beyond the dates from the report. We refer the reader to the supplementary figure and to text that we added in the methods section to clarify (Line 136):

(see Supplementary Figure 1 and Methods).

And Methods Section (Lines 901-906):

Note that SPARC modeled lifetime estimates begin in 1960 and end between 1998 and 2010, depending on the model. Because we require a lifetime estimate for all years between 1955 and 2016, we extend each model's initial values from 1960 to earlier lifetimes (i.e. from 1955-1959), and extend their end values to all subsequent years until 2016. The time-dependent lifetime is then taken to be the mean of these extended modeled lifetimes.

Line 144. DE is first defined here simply as direct emissions (which suggest units something like Gg/y), but it is not until line 449 that it is defined as a fraction. It would be helpful to clarify this earlier and possibly call it a direct emissions factor to differentiate it from absolute direct emissions.

We changed the wording when we introduce DE to call it a direct emissions factor (Line 231).

Lines 169-170. The posteriors (Fig. S5) look very different to me, especially for 1985.

This is true, especially for the 1980s. We modified the text to clarify (Line 264-267):

The most noticeable difference in posteriors between the two lifetime scenarios occurs in the 1980s where the SPARC MMM lifetime results in lower production posterior than the constant lifetime scenario of 45 yrs.

Lines 183-184. What impact does the time dependency of the SPARC MMM lifetime have on this? Might the divergence after 1990 be driven by the fact that the SPARC MMM time dependent lifetime reduces over the period 1960 to 2006 such that it exceeds the mean value of 62.9 years in the earlier years and in the later years is less than 62.9 years (Fig. S1). Hence, after 1990 when emissions are strongly related to release from the banks, the SPARC MMM run with a lifetime of <62.9 years at that time will lead to greater emissions and so leading to a smaller bank than a run with a fixed lifetime of 62.9 years. The different posterior release fractions would then be a reflection of this.

Yes! This is a nice way of framing the impact of the time-dependent scenario. We modified the text to make this point (Line 293-302):

This divergence is driven largely by the fact that the SPARC MMM lifetimes are decreasing throughout the time period such that prior to 1980, the SPARC MMM is larger than 62.9 years, and from 1981 onwards, it is smaller. In recent decades, when emissions are strongly correlated with RF, the constant lifetime scenario results in lower RF posteriors and thus smaller reductions in bank size relative to the time-dependent scenario. Because RF has high temporal correlation, the constant lifetime scenario used here has a consistently lower RF throughout. Prior to 1980, when the constant lifetime is lower than the SPARC MMM, differences in production compensate for lower RFs, producing similar bank sizes between the two scenarios.

Line 198. It would be informative to state the average value of the SPARC MMM lifetime and to refer to Fig. S1.

This has been added to the text (Line 312-325):

This similarity in bank size occurs because the SPARC MMM lifetime has an averaged lifetime of 101.5 years over the period where observations are available (i.e. 1980-2016, see Fig S1), which is close to the constant lifetime estimate of 100 years for CFC-12.

Line 201. It would again be informative to state the average value of the SPARC MMM lifetime and to refer to Fig. S1.

This has been added to the text (Line 328-329):

(an average lifetime of 98 years from 1980-2016 for the SPARC MMM scenario versus a constant lifetime of 80 years, see Fig S1),

Line 203. Fig. S6 shows $\sigma \times UB$ (emissions (Gg)). The values for CFC-113 are similar or slightly less than for the other gases. However, because the emissions of CFC-113 are less, this implies that its values of σ must be larger than for the other CFCs. Presenting a figure of σ only might illustrate this point better.

The uncertainty distribution has an upper bound of $\sigma \times UB$ (we corrected this in the main text, Line 233), where UB is either 40 Gg or twice the value of top-down emissions uncertainty due to lifetimes, whichever is larger (see Methods, Line 1062-1063). For CFC-113, 40 Gg is used as the upper bound. The distribution of σ is the same for each gas so you wouldn't see the relative differences plotting σ alone. To illustrate the point that the reviewer is suggesting, we added a row in Supplementary Figure 9 (previously Figure S6) that shows ($\sigma \times UB$) relative to mean emissions over the simulation period. Here we can see that uncertainties relative to emissions are about twice as large for CFC-113 than CFC-11 or 12.

Line 207-221. The Fig. 1 caption needs some improvement. Please state what the Ashford, TEAP, WMO and IPCC/TEAP markers and lines represent, noting that they are bottom-up as this is how they are referred to in the text. The titles given to the plots whilst broadly correct could be improved for consistency of plots. Bank estimates or bank size. Plot (c) isn't just BPE. I suggest replacing "the production prior includes an unexpected emissions scenario" (Line 216-217) with "the production prior includes additional production to account for unexpected emissions". The red line is not the only difference (Line 218) as (d) has the red dashed and blue dashed lines from (a) instead of the black line in (c). There is also no line for a run with unexpected emissions. Unlike (e) (Line 220) red dashed and blue dashed lines are not in (c). The unexpected emissions scenarios (Line 220) needs to be mentioned for (d) as well. Also, it would be

good to mention the lifetimes used for CFC-12 and CFC113, both the fixed one and the average value of SPARC MMM LT noting that it is time varying.

We rewrote the caption for Figure 1 to address these issues (Line 334-425):

Comparison of banks derived from Bayesian Parameter Estimation (BPE) along with previously published values, top-down bank estimates, and alternative assumptions. a) Top-down CFC-11 bank estimates assuming known lifetimes and reported production (see eq 2). Banks are derived using SPARC multi-model mean (MMM) time varying atmospheric lifetimes (blue) and a constant lifetime of 45 years (red). (b) Top-down CFC-11 bank estimates assuming SPARC MMM time varying lifetimes and three production scenarios: Reported production (blue), $1.05 \times$ reported production (red), and $1.1 \times$ reported production (yellow). For (a) and (b) production values are based on AFEAS and UNEP reported values (see Methods). (c) BPE derived CFC-11 bank estimates assuming the SPARC MMM lifetime (blue) and constant lifetime of 45 years (red). The grey line is analogous to the blue line but production prior includes additional production to account for unexpected emissions from 2000-2016 (see Methods). (d) BPE derived CFC-11 bank estimates assuming SPARC MMM time varying lifetimes (average value of 62.9 years) shown in blue, and constant lifetime of 62.9 years is shown in red. Dashed lines are corresponding top-down bank estimates. (e) BPE derived CFC-12 bank estimate assuming SPARC MMM lifetimes (average value of 112.9 years) shown in blue, and 100-year lifetime is shown in red. Dashed lines are corresponding top-down bank estimates. (f) BPE derived CFC-113 bank estimates assuming SPARC MMM lifetimes (average value of 106.3 years) shown in blue, and 80-year lifetime is shown in red. Dashed lines are corresponding top-down bank estimates. The black line in (a) - (c), (d) and (f) is the WMO (2003) bank estimate. For (c) – (f), the BPE median estimates are shown using thin solid lines with the 95% confidence intervals indicated by corresponding shaded region. The markers in plots (c) and (e) indicate previously published bank estimates as follows: the green marker is from Ashford (2004)⁹, the red marker is from TEAP(2009)¹⁵, the black marker is from WMO(2018)³, where banks were derived beginning with TEAP(2009)¹⁵ estimates, and the pink marker is from TEAP (2019)¹⁶.

Lines 253-255. The Bayesian analysis seems to suggest that emissions of CFC-12 were larger than the observationally-derived emissions from around the mid-1990s to ~2006. What are the implications of this? Interestingly, during this same period, the observationally-derived emissions for CFC-11 are at the lower boundary of the Bayesian confidence interval (or even below for the fixed lifetime). Could this be related to what is seen for CFC-12?

This behavior is interesting, but we don't have enough information to analyze it further. It is possible that a change in atmospheric circulation occurred that would affect both lifetimes, but other explanations are plausible. For example, it is also possible that both gases were subject to stockpiling (e.g., late 1980s-early 1990s) as the phaseout occurred, but we don't have information to quantify this. We added a few sentences to this effect on line 468-473:

It is interesting that the observationally-derived emissions for both CFC-11 and CFC-12 lie at the lower edge of the Bayesian estimates from the mid

1990s to mid-2000s. Potential reasons for this behavior could include transient changes in circulation and hence lifetimes of both, or releases from stockpiles of both as phaseouts occurred, but other explanations such as larger errors in production are also possible.

Line 260. Fig. 3. I think it would be informative to add a line for the scenario which includes unexpected emissions in the inset in (a).

We agree that a comparison with the unexpected emissions scenario would be informative. Because the unexpected emissions scenario includes a posterior distribution, we thought it was too much to add to Figure 3(a). Instead, we have added an analogous plot to Figure 3a to the supplement (Fig S11) where the Bayesian model is run including unexpected production in its prior. We add a reference to it in the main text (Lines 463-465):

When unexpected production is accounted for in prior production, the posterior emission space encompasses observationally-derived emissions (see Figure S11).

Line 295. “implying ongoing contributions to ozone destruction in the future even if further production ceases now.”. This could be expanded to clarify that this ongoing contribution is in addition to the on-going contribution that we already expected from what had previously been thought to be in the bank.

This has been expanded to clarify (Lines 530-532):
implying ongoing additional contributions to ozone destruction in the future beyond those previously thought to be in the bank, even if further production ceases now.

Line 306. Full stop before “Global”.
This has been corrected. (Line 546)

Line 313. Subscript for CO₂.
This has been corrected (Line 561)

Line 316. Fig. 4 states “median” as opposed to “mean”. Which is it?
Thank you - It is median. This has been corrected (Line 564).

Lines 319-321. It was not immediately clear to me that Scenario 2 is the same as the idealized best case. Suggest "In Scenario 2 we consider an idealized best case in which there is 100% recovery and destruction of the CFC banks in 2020 and no further emissions past 2020."

This wording has been adopted (Line 568- 570)

Lines 327-328. It would be helpful to clarify in the text that the source of the EESC data in Fig 4 (left (a)) is from the WMO 2018 Assessment. Lines 364-365 state that EESC values leading up to the scenarios are from the WMO, but am I correct in that the mixing ratios for all chemical species other than the CFCs come WMO, right up to 2100? Please clarify.

We edited the text to clarify (Lines 575-577):

With the exception of CFCs, EESC values use mixing ratios from the WMO 2018 Assessment. For CFCs, EESC values are estimated using mixing ratios from the WMO 2018 Assessment leading up to the scenarios.

Lines 332-334. This comment makes a comparison between mixing ratios 2000 to present with or without banks being destroyed. Surely this is the comparison between Scenarios 1 and 2 which are illustrated in Fig 4 (right (b)). I would therefore suggest moving this comment to after where Fig. 4 (right) is described in the text.

We moved this comparison as suggested (Line 606-610):

While Figure 4 (a) illustrates that CFCs have declined slightly from 2000 to present, the ongoing emission from banks (even without additional unexpected emissions) means that they have contributed less to the total reduction in EESC than they would have if the banks had been destroyed (e.g. Scenario 3 vs Scenario 2).

Lines 355-365. Fig. 4. It would be helpful to have tickmarks on the x-axis. In the text the plots are referred to as left and right, but in the caption as (a) and (b). Please be consistent. I appreciate that there is a superscript pointing to the WMO 2018 Assessment for plot (a), but it would be helpful at this point to state the source as done in line 365. Is it the mean or median RF and bank sizes? The text refers to the mean RF.

- We added tick marks and added (a) and (b) to the figure.
- We changed the text to be (a) and (b) instead of (left) and (right)
- We stated that the mixing ratios were from WMO2018 at the superscript
- We corrected the main text to median which is now consistent with the caption

Lines 374-378. This point about the difference between WMO 2018 EESC and the scenario 1 EESC is illustrated in Fig. 4. I suggest it would have been better to have made this point earlier when discussing Fig. 4. This would then separate it from the new point being made here which is that the unexpected emissions presented in Fig. 5 as Scen 2 extend the recovery by 1 year, from 2084 to 2085. It would help to refer to the scenario number in the text (see also next comment below). Please also clarify if the EESC plotted in Fig. 4 is "Antarctic" Cl loading as referred to here.

For figure 4, we are trying to make the point that different policies on bank destruction can have large impacts for ozone recovery. In section discussing Fig 5, we are

illustrating how the different assumptions regarding production, and different methods for estimating bank size (i.e. the BPE method vs WMO's method) lead to large differences in estimate ozone recovery time. So – we chose to keep the WMO comparison where it is.

We have clarified that EESC is Antarctic CI throughout (Lines 561, 611, and 628)

Lines 386-396. Fig. 5. Please explain in the caption what is represented by the grey shading around the black lines. I couldn't find mention of Scen 3 in the text, nor could I find mention of Scen 2 as it relates to CFC-113 (plot (c)) in the text. It was not easy to follow the different scenarios that were performed since those presented in Fig. 4 are called Scenario 1, 2 and 3, whilst those in Fig. 5 are called Scen 1, 2 and 3, and it appears that some of the scenarios with the same number are different between the figures and even between plots in the same figure. It would really help to have a set of unique scenario numbers and to ensure that all scenarios are discussed in the text, not just the figure captions.

We clarified what the grey region indicates (Line 693-694):

The shaded grey region represents 1-s.d. of uncertainty due to uncertainties in bank estimates.

We also changed the name of these scenarios to A, B, and C so as not to be confused with the other scenarios described in the main text.

Lines 421-422. What is the source of the production data used in the top-down estimates presented in this paper?

We only estimate banks using the top-down method as an illustrative example in Figures 1(a) and (b). Here we show how sensitive the banks are to differences in lifetimes and production. We added clarifying language to the caption where we illustrate the sensitivity of the top-down method to differences in production (Line 340-341)

For (a) and (b) production values are based on AFEAS and UNEP reported values (see Methods for details).

Line 489. "to apply □□?"

This has been corrected.

Line 536-538. How were the SPARC time dependent lifetimes extended to 1955 and 2016?

We clarified in the Methods Section (Lines 901-906):

Note that SPARC modeled lifetimes begin in 1960 and end between 1998 and 2010, depending on the model. Because we require a lifetime estimate for all

years between 1955 and 2016, we extend each model's initial values from 1960 to earlier lifetimes (i.e. from 1955-1959), and extend their end values to all subsequent years until 2016. The time-dependent lifetime is then taken to be the mean of these extended modeled lifetimes.

In the equations used in the Methods section and the Supplementary Material, I would recommend using the same names for production, either Production or Prod, unless these two terms are meant to represent something different, in which case please clarify.

This has been changed to be Prod throughout

Equation 3. Whilst this equation is correct wouldn't a more general equation to describe the bottom-up approach be to calculate the integral from in year y1 to t, similar to equation 2?

We agree that an integral would be a cleaner comparison to equation 2. However, we find the integral expression (shown below) is not as intuitive for readers so we prefer to keep it in this form when it is first introduced. We have edited the text to clarify how banks are modeled in the BPE and add the integral equation in Lines 822-826:

The prior distributions for Bank input parameters are not independently defined. Instead they are simulated as a function of prior distributions for all previous timesteps of RF, DE, and production. They can be estimated by iterating eq(3) forward in time, or equivalently;

$$Bank_t = (1 - DE_t)Prod_t + \sum_{y=y1+1}^{t-1} (1 - DE_y)Prod_y \prod_{j=0}^{t-y-1} (1 - RF_{t-j}) + Bank_{y1} \prod_{j=y1+1}^t (1 - RF_j),$$

where y1 is the first year in the simulated time period.

In several places throughout the methods it is stated that Bank_{t-1} is one of the input parameters. I struggled to understand how a prior can be used for Bank_{t-1} as it would be time varying, dependent on how much has been released from the bank ((1 - RF_t)*Bank_{t-1}) and how much new production has gone into the bank ((1 - DE_t)*Production_t). This was compounded by the description of the Priors for Input Parameters only describing production, RF and DE. Also, in Supplementary Material plots of the priors are only given for the production, RF and DE (Figs. S8, S9 and S10) and the headings only refer to Direct Emissions, Release Fractions and Production and not banks. I can see from lines 637-638 that the RFs depend on the composition of the bank so imply priors for the bank. Further description of the bank priors would be useful.

We think that our clarifying text in response to the previous comment addresses some of this confusion around the bank priors as well. We also added clarifying text in the Methods Section when describing how RF and DE are derived (Line 1032-1037):

Also note that for total production, we use AFEAS data up until 1989, after which we use UNEP data. For estimating RF and DE, production data from AFEAS is used only to approximate relative production by equipment type over time. This, in turn, provides a prior estimate of the relative distribution of equipment type in the bank, which we use to estimate RF and DE. These RF and DE priors are constructed independently of total production priors.

Supplementary Material

It is not clear to me why $EF_{k,y1}$ and $EF_{k,bank}$ are used on page 2 rather than $DE_{k,y1}$ and $RF_{k,bank}$.

We wanted to distinguish between release fractions that are specific to an equipment type versus release fractions that are specific to the gas (i.e. aggregated across equipment type). We added clarifying text in the Supplementary Methods:

We use the term $EF_{k,bank}$ to refer to the release fraction specific to equipment type and RF to refer to the release fraction of a specific gas, which aggregates across all types of equipment for that gas. $EF_{k,y1}$ is the RF in the year of production, which we term Direct Emissions (DE) when aggregated across all equipment types.

Figs. S2, S3, S5. Presumably for CFC-11? Suggest using the same ranges for the scales of the axes to help comparison. What is the constant lifetime value, 45 years? Why are the priors different for the two different lifetime scenarios?

The prior distributions appeared different across lifetime scenarios due to different bin sizes and a low sample size used to create the figure. We fixed bin size across scenarios and drew a larger sample from prior and posteriors distributions to create these figures. We also clarified that this is for CFC-11, and that the constant lifetime value is 45 yrs. We fixed the range of the axes so that they are the same across lifetime scenarios for comparison.

Fig. S4. Presumably for CFC-11? Units of production? Which BPE run (e.g. lifetime)?

We corrected these figures to add the units of production and clarified that this is for the CFC-11 SPARC MMM lifetime run.

Reviewer #3 (Remarks to the Author):

What are the major claims of the paper?

This paper utilizes Bayesian parameter estimation (BPE) to assess the uncertainty in CFC-11, CFC-12 and CFC-113 banks and bank emissions. The underlying models and distributions were built using expert knowledge available to the authors from a variety of sources, and the authors acknowledged and dealt with known and potential biases and inconsistencies in these sources. Interpretation of the posterior distributions and comparison of the posterior distributions to existing non-BPE approaches as well as observed emissions data, provided conclusions regarding potential delays to ozone hole recovery and questions regarding sources of CFCs. The authors illustrate how BPE can be used to provide better measures of the uncertainty in conclusions than can be provided by non-probabilistic approaches.

Are they novel and will they be of interest to others in the community and the wider field?

The application of BPE to CFC emissions is to my knowledge novel and extends the use of BPE to a new area. The authors make the case that the use of BPE in modeling emissions and banks better characterizes the uncertainties we have and could lead to different decision make processes in the real-world.

* Is the work convincing, and if not, what further evidence would be required to strengthen the conclusions? Comment on the appropriateness and validity of any statistical analysis.

Did the authors examine the distribution of the joint prior (or model induced outputs using jargon of Bates et al (2003) and Poole and Raftery (2000)) distribution of RF and Banks for the years 1995 and 2005? The joint RF and Banks posterior distributions for these years shown in Figure S4, would appear to favor a narrow ridge of values. Why do we think this is?

We added the joint prior distributions for RF and banks, Prod and Banks, DE and banks, and RF and DE to the supplement (Fig S4-S7). The joint prior for RF and banks still shows a ridge, but the ridge for the prior is wider than in the posterior. We would expect a narrow ridge in the prior and posterior for later time periods for two different reasons: 1) For the simulation model, a low RF would lead to a larger accumulation in the banks leading up to 2005. Because RF has high autocorrelation, a low RF in 2005 would be correlated with a low RF throughout. This is why we see the strong negative correlation between RF and Banks in the prior. 2) For the posterior, emissions in later time periods (e.g. 2005) are largely a function of Bank size and RF ($emissions \cong RF \times Bank$ for later years). Therefore, we would expect values on the ridge where $RF \times Bank$ are closer to the observationally-derived emissions to have a higher likelihood than values further from the ridge. We added some text to explain this further on Lines 253-258:

This strong negative correlation between bank size and RF in recent decades is to be expected for two reasons: 1) For the simulation model, a low RF would lead to a larger accumulation in the banks in earlier

decades. Because RF has high autocorrelation, a low RF in earlier decades would be correlated with a low RF in recent decades as well, thus explaining the strong negative correlation between RF and Banks in the prior. 2) For the posterior, if the near-zero reported production in recent decades is accurate, then emissions must be entirely driven by the depletion of the banks, and thus controlled by RF (i.e. Emissions $\cong RF \times Bank$). Therefore, we would expect values on the ridge where $RF \times Bank$ are closer to the observationally-derived emissions to have a higher likelihood than values further from the ridge.

The authors describe the correlation of RF and DE and both are involved in the model for Banks, however it is not obvious that the posterior should be this restricted. Figures S2 and S3 shows that the posterior distributions for RF and DE in 2005 consisted of values resampled from the lower end of prior distributions for these parameters in that year. (This effect is stronger for RF than DE.) Is this seen in the joint prior also? I.e. did the joint posterior consist of resampled values along the boundary of the prior distribution? If so, did the authors examine sensitivity of the final interpretations to widening this joint prior?

These are great suggestions! We have added the joint prior and posterior for RF and DE to the supplement (S7). The joint posterior consists of resampled values close to the lower boundary for RF, but not DE. RF is physically constrained to being larger than zero and based on the uses of CFC-11, it would be unrealistic to be smaller than 0.01, so we don't think that widening the prior further would lead to a more realistic posterior. However, to test the reviewer's suggestion, we reran the model by widening the prior on RF value for closed cell foams for CFC-11, which is the biggest component of the bank in the last two decades of the simulation. The results are now included in the supplement (Fig S12). If the uncertainty in closed cell foams is doubled, this does impact the bank estimates, in particular the median and upper bound of the bank estimates, suggesting that our results are conservative given our choice of informative priors for RF. We have added a description of these sensitivities to the main text (Lines 476-488), which we have included in response to the next comment as well.

Did the authors examine sensitivity of the results discussed in the paper to minor perturbations in the parameters of the prior and hyperprior distributions? E.g. if such parameters were adjusted by 10%, what were the resulting changes in the posterior results? This is particularly important for parameters that are not well known.

This is another excellent suggestion! We reran the BPE to test for perturbations in Production, DE, and RF, uncertainty in RF, and the uncertainty range in correlation parameters. We have found that model results are insensitive to perturbations in DE and RF mean and moderately sensitivity to perturbations in RF variance and production. We have added figures of the first four of these simulations to the supplement (Fig S12) and have added some explanatory text to the main manuscript (increasing uncertainty range in correlation had no impact of final results): (Line 476-488)

Note that the results of the BPE analysis are constrained by our choice of priors, which have been developed using published estimates of the input parameters. We investigate the sensitivity of our results to various input parameters. In particular we test the sensitivity of bank size to ~10% increases in the mean of the prior distributions of RF, DE, and production as well as increases in the standard deviation of the RF prior distribution. For RF and DE, this is done by changing values of the loss rates for each equipment type in the bottom-up accounting model (see Methods and SI for details). We find that BPE-derived bank estimates are moderately sensitive to production values and RF uncertainties. Production is not likely to be lower than the reported values, which were used to construct the base case scenario, and the lower bound of RF is fairly constrained, implying that our choice of priors are likely leading to conservative estimates in the size of banks (see Fig S12).

* On a more subjective note, do you feel that the paper will influence thinking in the field?

I would like to hope that this paper makes a convincing enough argument to influence thinking in the field, specifically the questions it raises about the sources of CFC-113 and potential changes to the management of other CFC banks. BPE is, as the authors point out, useful in incorporating information from observed data as well as expert knowledge of the systems (through priors).

* Ability of a researcher to reproduce the work, given the level of detail provided.

The manuscript is written clearly with sufficient detail to encourage replication of the results. Supplementary materials are useful and relevant to the discussions. Replication would require access to any dataset that was not specifically enumerated in the materials.

Congratulations on a nice application of the method to a novel application. I particularly appreciated the grounding of conclusions and decisions in expert knowledge of the observed phenomena. Addressing the questions raised above regarding sensitivity analysis and possible effect of the prior distribution choices, would strengthen the paper and apprehension that some might have to the use of this BPE approach.

Thank you!

REVIEWERS' COMMENTS:

Reviewer #2 (Remarks to the Author):

The authors have done a very thorough job addressing all the reviewers' comments. In most instances they have followed the suggestions made, and in the few cases where they have not, they have provided explanation and clarification.

I am very happy to recommend this manuscript for publication.

Reviewer #3 (Remarks to the Author):

Thank you for addressing my questions regarding sensitivity analysis and the effect of prior distribution choices. I believe that the additions to the paper to address these questions, have added strength, and could alleviate apprehension that some may have to the use of the BPE approach.